

# Design and evaluation of a low-cost sensor node for near-background methane measurement

Daniel Furuta[1], Bruce Wilson[1], Albert A. Presto[2], Jiayu Li[3]

[1] Department of Bioproducts and Biosystems Engineering, University of Minnesota, 1390 Eckles Ave., St. Paul, MN 55108, USA
[2] Department of Mechanical Engineering, Carnegie Mellon University, 5000 Forbes Ave., Pittsburgh, PA 15213, USA
[3] Department of Mechanical and Aerospace Engineering, University of Miami, 1251 Memorial Drive, Coral Gables, FL 33146, USA

*Correspondence to*: Jiayu Li (jiayuli@miami.edu)

**Abstract.** We developed a low-cost methane sensing node incorporating two metal oxide (MOx) sensors, humidity and temperature sensing, data storage, and telemetry. We deployed the prototype sensor alongside a reference methane analyzer in two sites - one outdoors, one indoors - for several months each of data collection across a range of environmental conditions and methane levels. We explored calibration models to investigate the performance of our system and its suitability for background monitoring and enhancement detection. We achieved moderate performance in the 2 to 10 ppm range, but found that the sensor response varied over time, possibly as the result of changes in non-targeted gas concentrations. We suggest that this cross sensitivity may be responsible for mixed results in previous similar studies. We discuss the implications of our results for the use of these and similar inexpensive MOx sensors for near-background methane monitoring.

## 1 Introduction

With the well-known importance of methane emissions to climate change and the scarcity of high-resolution ground-level data, scientists and engineers are working to develop low-cost sensor networks and monitoring methods. Researchers and commercial entities have explored a variety of sensor mechanisms, including optical, pyroelectric, and chemiresistive devices, among others (Aldhafeeri et al., 2020). Due to their low cost, metal oxide semiconductor (MOx) sensor elements are appealing candidates for inexpensive sensor network development (Cho et al., 2022).

MOx sensor implementation poses a variety of technical challenges, and laboratory calibrations may not translate to real-world applications (Barsan et al., 2007). In particular, environmental factors such as humidity and interfering gasses are difficult to incorporate in a lab setting (Wang et al., 2010). Poor selectivity for target gasses is a challenge of particular relevance to our current study, and is a well-known problem with a variety of possible solutions including sensor array or "e-nose" configurations (Ponzoni et al., 2017; Cheng et al., 2021).



A variety of previous studies have explored the TGS MOx sensor product line from Figaro Engineering for methane detection. In particular, the TGS2600 and TGS2611-E00 sensors have promising reports in the literature (Collier-Oxandale et al., 2018; van den Bossche et al., 2017). The TGS2611-E00 is marketed for methane detection in alarms, leak monitoring, and similar applications (Figaro USA, Inc., 2013); the TGS2600 is marketed as a general-purpose air contaminant sensor for use in air

quality monitors and similar devices (Figaro USA, Inc., 2022). Both are low-cost sensing elements available for around $20 each from electronic component distributors (Maritex Co., September 2023 prices).

Previous work has found TGS2611-E00 to perform reasonably well when calibrated in a laboratory setting, with error within ±1.7 ppm across a 2 to 9 ppm methane concentration (van den Bossche et al., 2017). Several field experiments have found

useful performance for detecting methane concentrations in a higher range; among them, Cho et al. (2022) find TGS2611-E00 effective above 100 ppm concentrations, and Jørgensen et al. (2020) report success in the 2 to 100 ppm range. Riddick et al. (2022) successfully detected large changes in methane concentrations, corresponding to natural gas leaks, but found that quantitative emission estimates had poor accuracy. Shah et al. (2023) provide an in-depth examination of TGS2611-E00 calibration, including the important observation that laboratory calibrations may not generalize to different ambient conditions

in the field. All of the previous studies note that environmental conditions, particularly humidity levels, affect sensor response; Shah et al. (2023) also provide some evidence that ambient gas makeup plays a significant role in sensor behavior.

TGS2600 has also found positive results in some studies. Eugster and Kling (2012) report a general sensor correspondence with methane trends in a field study, although with a low $R^2$ of 0.19. Other papers find better performance (Collier-Oxandale

et al., 2018; Riddick et al., 2020; Eugster et al., 2020; Casey et al., 2019), but generally with complicated algorithms required, and in some cases with notable differences in performance between laboratory and field settings (Riddick et al., 2020) or from site to site (Collier-Oxandale et al., 2018).

The previously mentioned studies explore a range of methane concentrations. We focus in this paper on a low concentration

range, which we define as ranging from the atmospheric background of approximately 2 ppm to 10 ppm. Our previous laboratory work suggests that TGS2611-E00 has some methane response in this range and that TGS2600 does not (Furuta et al., 2022). It is unclear from previously published work whether these sensors are viable for monitoring the 2 to 10 ppm range in real-world settings, and whether these sensors are therefore usable in low-cost sensing networks for monitoring small fugitive emissions and similar low-concentration applications.


To better understand the viability of low-cost, MOx sensors for monitoring methane in this low concentration range, we designed an inexpensive sensor node complete with telemetry and datalogging, deployed the node in an outdoor setting and an indoor setting with a range of methane concentrations for several months each, and then characterized the sensor response to both environmental conditions and methane levels. We present the full design for our sensor node and mention its strengths





and shortcomings. We highlight several difficulties in monitoring this low methane concentration range with MOx sensors, and discuss the suitability of the sensing approach for the concentration range of interest.

## 2 Methods

### 2.1 Sensing system design

We designed and built a system consisting of two MOx sensors, a relative humidity and temperature sensor, data storage and
telemetry, and power supplies and interfacing circuitry. We implemented the system on two printed circuit boards (PCBs), with one PCB holding the sensors and associated circuitry and the other PCB holding the data storage and telemetry components, as described in full in Appendix A.

We chose TGS2611-E00 (Figaro USA, Inc., 2013) and TGS2600 (Figaro USA, Inc., 2021) MOx sensors from Figaro
Engineering Inc. as our primary sensing elements. Based on our previous work, we expected TGS2611-E00 to respond reasonably well to methane and TGS2600 to show a weak or no response to methane in the concentration range of interest (Furuta et al., 2022). By including both sensors we hoped to allow our calibration algorithm to compensate for non-methane interfering gasses that might be detected by both TGS2600 and TGS2611-E00.

MOx sensors require stable power supplies for accurate readings, with previous studies noting that power supply fluctuations compromise performance (van den Bossche et al., 2017; Shah et al., 2023). Our overall system ran from a 5V DC power supply; to ensure high stability for the sensing elements we operated them at 4.8V derived from an onboard precision voltage regulator. We burned in the sensors and regulator for a week prior to data collection.

MOx sensors vary in resistance in response to target gasses. To convert this variation to a voltage signal that could be easily digitized, we placed the sensors in voltage divider configurations against selected reference resistors. For maximum sensitivity, the reference resistance value should be close to the expected sensor resistance; we used results from our previous work to choose reference resistor values of 15 k$\Omega$ for TGS2600 and 75 k$\Omega$ for TGS2611-E00 (Furuta et al., 2022). We digitized the voltage output for each divider using an ADS1115 16-bit analog to digital converter (Texas Instruments, Inc.).

To confirm power supply stability, we also digitized the sensor power supply voltage. Our system performed well in both sections of the experiment, with 95% of all data showing a sensor supply voltage within ±0.25 mV of the mean and 99.99% of all data showing a supply voltage within ±0.80 mV of the mean across the full dataset.

We recorded relative humidity and temperature using an SHT31-DIS sensor (Sensirion AG), which produces digitized readings with 2% relative humidity accuracy and 0.2°C temperature accuracy.



An M0 Adalogger microcontroller module (Adafruit Industries) controlled the overall system and recorded readings every five to six seconds to an SD card. We used a Boron microcontroller module (Particle Industries, Inc.) as a cellular modem for
telemetry; the Boron module also provided accurate timestamps for the datalogger. We found one brief period of corrupted readings, likely due to SD card malfunctioning, which we removed from the dataset. We did not note any other obvious problems with the system functioning.

We mounted the electronics inside a small enclosure with holes cut to allow air entry, seen in Fig. 1A. We screened the opening
to prevent debris, insects or other objects from entering the case. Our system operated passively, without an air pump.

We provide full schematics and a detailed electronic description of the sensing system in Appendix A.

## 2.2 Experimental design

We characterized the sensing system at two different sites, with a co-located LI-7810 methane analyzer (LI-COR, Inc.) as a
reference device. The LI-7810 monitors methane, water vapor, and $CO_2$ using optical spectroscopy with precision better than 1 ppb for methane and 50 ppm for water vapor, with a frequency of one reading per second (LI-COR Inc., 2023). We did not use the $CO_2$ measurements for this paper. Figure 1 depicts the experimental sites and setup.



Figure 1: **A shows the sensor node construction, which is described fully in Appendix A. B is the indoor study site with the sensor node and reference analyzer placed close to a demonstration anaerobic digester. C is the outdoor study site, with D showing the positioning of the sensor node and reference analyzer intake.**

### 2.2.1 Outdoor site: ambient levels with short controlled releases

Our first site was an urban yard in Minneapolis, USA without any known notable methane sources nearby; our intention for this site was to characterize our sensor's performance close to background methane levels. We located our sensing system and the reference analyzer's sample intake near the house exterior, as seen in Fig. 1C and D. The reference analyzer drew samples through approximately three meters of tubing: as we later averaged the data to a ten-minute timescale, we considered the resulting sampling lag of less than 30 seconds to be negligible for the purpose of our analysis.





The background methane concentration at our research site was approximately 2 ppm, with a minimum observation of 1.98 ppm. We had no expectation of elevated levels. To provide some range of methane concentrations, we performed a small number of controlled releases in the vicinity of our sensor setup from a 2.5% methane gas cylinder. In August 2022 we released methane eight times through a 2 L/min regulator for 30 seconds each and once for 15 seconds. In October and November 2022 we released methane 40 times through a 0.1 L/min regulator for 10 minutes each. These releases produced a maximum 10-

minute averaged methane value of 5.8 ppm, with most of the releases producing methane concentrations between 3 and 4.5 ppm.

### 2.2.2 Indoor site: methane leaks and elevated levels

Our second site was indoors in the Biosystems Engineering building at the University of Minnesota, Twin Cities campus. We placed our sensing system and reference analyzer close to a benchtop classroom-scale demonstration anaerobic digester, as

shown in Fig. 1B. The digester was located in a large workshop area, and we expected a variety of methane levels resulting from small leaks or larger pulses when biomass was added to the system or when methane was removed. We also expected possible emissions of methane and other gasses from surrounding labs, some of which were working on fermentation and bioprocessing projects. Since this site was indoors, we expected a narrower range of temperatures and humidities than at our outdoor site.


We collected data from the outside site from July to November 2022 and from the indoors site from January to May 2023.

### 2.3 Data processing

The reference analyzer showed clock drift over the sampling periods, which we noted and corrected. We converted the raw MOx sensor readings to resistance values using the recorded supply voltages and known voltage divider resistor values. We

associated the reference and sensor data by their timestamps.

The indoor portion of data collection saw several large, short methane spikes which exceeded the specified range of the reference analyzer; as we were unable to guarantee data quality in these periods, we dropped all data within a one-hour window of a methane value exceeding the reference analyzer upper limit of 100 ppm. We chose the relatively long window to ensure

that both the reference analyzer and sensor node had returned to baseline conditions after each spike. The outside experiment lost records from the reference analyzer for two weeks, leading to a gap in the data. Our sensor experienced power failures on several occasions through the experiment, as well as sporadic downtime to allow for data retrieval. As the MOx sensors have internal heaters which require some time to stabilize, we removed two hours of data after each reboot.



To reduce the effect of any lag between our passive sensor node and the active-sampling reference analyzer, we chose to perform our analysis on the dataset averaged to a 10-minute timescale.

Our research interest is the low methane concentration range, which we define as background to 10 ppm, with a background concentration of approximately 2 ppm in our location. We removed averaged data with concentrations above 10 ppm as well

as the data immediately following; this resulted in removing 31 of 15688 records for the inside set, and 3 of 14333 records for the outside set.

**2.4 Sensor calibration**

MOx sensors have well-known sensitivities to humidity and other environmental conditions. Previous studies, as well as manufacturer datasheets, have often related methane to the sensor response using a baseline value, which represents the

expected sensor response to environmental conditions other than methane (Figaro USA, Inc., 2013; Eugster and Kling, 2012; Riddick et al., 2022; Shah et al., 2023). Methane is then fit by an equation using the sensor deviation from the expected baseline value. We proceeded with this two-step approach as follows.

First, we estimated the baseline TGS2611-E00 sensor response without methane enhancements by fitting the response as a

function of environmental conditions for time periods where the measured methane concentration was below 2.3 ppm. We examined the effect of several environmental parameters: temperature, water vapor concentration, and elapsed sensor running time. We chose to include sensor run time to incorporate any effects of sensor aging; the time parameter also potentially included any effects from other environmental parameters we were not equipped to measure, such as interfering, non-target gasses.


Our experiment collected both water vapor concentrations and relative humidity data from the reference analyzer and a PCB-mounted sensor respectively. Relative humidity is dependent on water vapor concentrations and temperature, and our previous work found water vapor concentrations to predict sensor response better than relative humidity (Furuta et al., 2022); accordingly, we decided a priori to include water vapor concentrations and not relative humidity as a possible term in our

analysis.

After evaluating the possible regressions, we selected Equation 1 as the best performing fit for the TGS2611-E00 baseline across the inside, outside, and combined datasets, again fitting only data with methane concentrations below 2.3 ppm. We provide a detailed description of the equation selection process in Appendix B.

$\log(baseline) = \beta_0 + \beta_1 \log(H_2O) + \beta_2 T + \beta_3 \log(time)$              (1)





Our sensing system included a secondary sensor, the TGS2600, which we found previously not to respond to methane in the concentration range of interest (Furuta et al., 2022). We examined whether including this sensor in the baseline prediction could improve accuracy, most likely by accounting for non-target interfering gasses. Adding this sensor to Equation 1 produced

Equation 2 as a possibly improved candidate for the baseline regression.

$$\log(baseline) = \beta_0 + \beta_1 \log(H_2O) + \beta_2 T + \beta_3 \log(time) + \beta_4 \log(TGS2600) \tag{2}$$

The TGS2611-E00 sensor response is expected to deviate from the predicted baseline response as a result of methane levels as shown in Equation 3a. We rearrange this equation to predict methane levels in Equation 3b.

$$\frac{TGS2611}{baseline} = \alpha + \beta CH_4 \tag{3a}$$

$$CH_4 = \frac{1}{\beta}\frac{TGS2611}{baseline} - \frac{\alpha}{\beta} \tag{3b}$$

We used the combination of Equation 3 with either Equation 1 or Equation 2 to calibrate the TGS2611-E00 response to methane. We evaluated the performance of each component as well as examined challenges for the calibration method.

**3 Results**

**3.1 Collected data**

Figure 2 plots the time series of cleaned, 10-minute averaged data for the two experiments. As mentioned previously, data loss from the reference analyzer led to a two-week gap in late August and early September. Both experiments had occasional smaller gaps not visible on the plots due to power failures, system downtime for data collection, methane spikes exceeding the

reference analyzer specifications, and similar causes.

The outside experiment captured a range of temperatures and humidities over its 113-day span. The experiment began in summer, with associated high temperatures and water vapor concentrations; the end of the experiment was in autumn, with colder temperatures and lower water vapor levels. The temperature sensor was inside the system's enclosure, which

experienced consistently higher temperatures than the outside air.

Some fluctuations in methane concentrations are visible in the outside dataset. We observed a diurnal cycle in methane levels for some periods of the experiment, possibly due to soil processes near the sensor; we give an example of this cycle in Appendix C. Several unplanned sharp spikes occurred beyond our controlled releases, which may have been due to residential gas leaks

or similar events.



The inside experiment ran for 111 days, beginning in winter and ending in spring. During winter, the location was heated to a relatively consistent temperature without added humidity. Accordingly, the temperatures were more stable than in the outside experiment, with overall lower water vapor levels and relative humidities. As with the outside sensor, the recorded

temperatures were inside the sensor node enclosure, which was warmer than the ambient air.

**Figure 2: Time series of both experiments, with internal sensor node temperatures (A), water vapor concentration and relative humidity (B), methane data collected by the reference analyzer (C), and MOx sensor responses (D).**

Figure 3 shows data distributions and correlations between the different measured variables for the two portions of the

experiment. In the outside experiment, the TGS2611-E00 and TGS2600 sensors were highly correlated (Pearson's r=0.97), and both were strongly correlated with water vapor levels (|r|>0.8) and temperature (|r|>0.7). The sensors were not obviously correlated with methane levels (|r|≤0.14).



Sensor performance can be influenced by cumulative sensor runtime in a variety of ways, which we will examine in more
detail. We chose to capture this parameter as elapsed run time, representing the cumulative time the sensor had been powered
on since the beginning of the outside experiment.

As seen in Fig. 3, sensor response was strongly correlated with elapsed time (r>0.7) in both the inside and outside experiments.
Due to seasonal changes, elapsed time was itself correlated with both water vapor concentrations and temperatures (which
were themselves also correlated), as outdoor temperature and water vapor levels are higher in summer than in winter.
Accordingly, it is possible some portion of the sensor-time correlation was due to the well-known sensitivity to water vapor,
or due to an effect from temperature.

The MOx sensors were again strongly correlated with each other in the inside experiment (r=0.88). Sensor correlation with
humidity (|r|>0.5) was stronger than with temperature (|r|>0.3), and the sensors were correlated moderately with elapsed time
(|r|>0.5). Neither sensor showed a strong correlation with methane levels (|r|≤0.12).



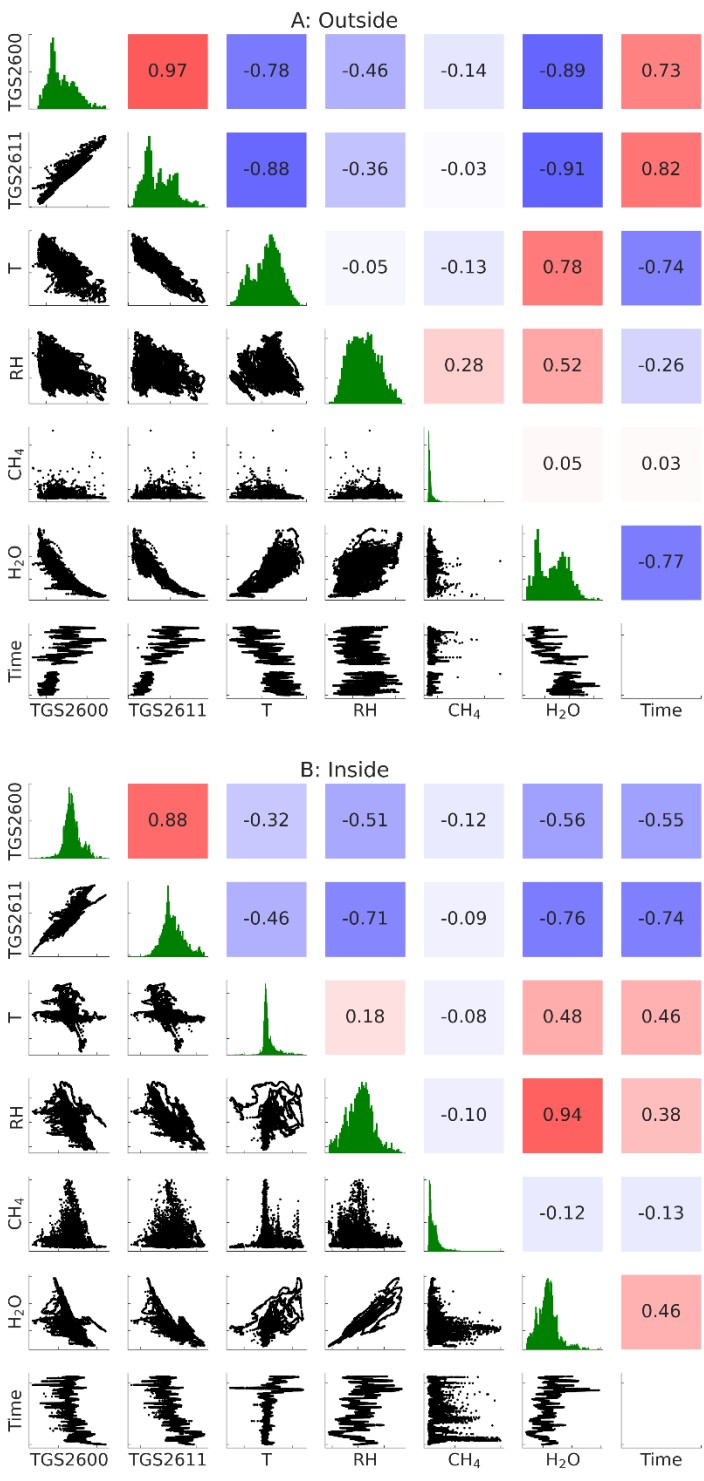

**Figure 3: Correlations and data distributions for the outside (A) and inside (B) datasets. The upper triangle shows correlations (Pearson's r); the lower triangle shows pair plots between variables; the diagonal line shows histograms for each variable.**





## 3.2 Baseline sensor response

As described previously, we examined the baseline TGS2611-E00 response to environmental conditions. Our experiment recorded four parameters with potential effect on sensor response besides methane: temperature, water vapor concentration, relative humidity, and time. We used water vapor concentration and ignored relative humidity a priori based on our previous work (Furuta et al., 2022). Our first candidate baseline equation used these environmental factors to predict TGS2611-E00.

Our second candidate equation included a secondary sensor, TGS2600, which we have previously found not to respond to methane at low concentrations; by including TGS2600 in the baseline response, we can possibly remove influence from non-target gasses and other unexpected factors. We selected all data with a measured methane concentration below 2.3 ppm, a value slightly above the background levels of approximately 2 ppm at our research locations, as the targets for our baseline regressions and compared the performance of the two baseline equations.

As previously described, we selected Equation 1 as the best performing regression without TGS2600. This regression closely fit the baseline TGS2611-E00 response, again defined as the sensor response for all datapoints with methane levels below 2.3 ppm. We obtained $R^2$ values of 0.97, 0.91, and 0.89 for the outside, inside, and combined datasets respectively, and root mean square error (RMSE) values of 1.46, 1.56, and 2.81 kΩ respectively.

The regressions capture the overall trend in the baseline response, as seen in Fig. 4A and C, but with considerable variance. As shown by the color-coding, time appears to have an important effect on sensor response, supporting the inclusion of time in Equation 1; for example, in the inside dataset in Fig. 4A, which had relatively little variation in humidity and temperature, the sensor shows a markedly different baseline response at different time periods of the experiment.

We then added the TGS2600 sensor response to the baseline equation as a potentially stronger regression. We have previously found that TGS2600 does not strongly respond to methane in the 10 ppm concentration range (Furuta et al., 2022). Accordingly, we hoped that the TGS2600 response would allow the baseline fit to accommodate unknown environmental effects, such as changes in ambient non-targeted gas concentrations, without affecting the relationship between the baseline fit and methane levels. Adding a term for TGS2600 produced Equation 2, which we again fit for the low-methane data subsets.

Adding the TGS2600 term improved $R^2$ to 0.99, 0.98, and 0.98, and RMSE to 0.71, 0.73, and 1.21 kΩ for the inside, outside and both datasets respectively. As seen in Fig. 4E and G, the regressions still show some error but the fit is closer than with Equation 1, suggesting that Equation 2 outperforms Equation 1 for predicting the TGS2611-E00 baseline response.





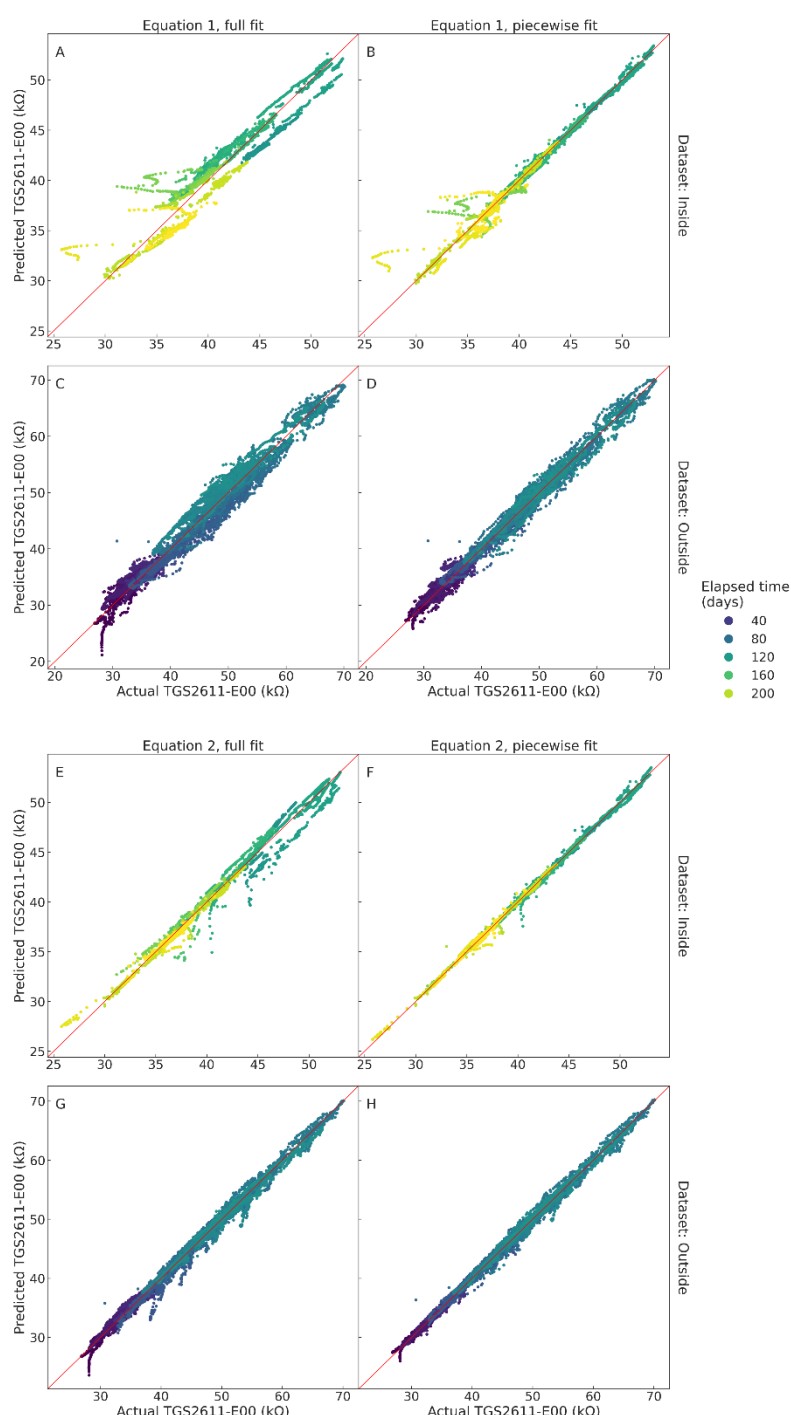

**Figure 4: Results of TGS2611-E00 baseline regressions for the inside (A, B, E, F) and outside (C, D, G, H) datasets, filtered to only include data with 2.3 ppm or less of methane. The system run time from the beginning of the outside experiment in days is coded as color. A-D shows the fit for Equation 1, and E-H for Equation 2, both with regressions over the full filtered datasets (A, C, E, G) and piecewise by time (B, D, F, H).**





Even with the additional sensor term, the accuracy of the regressions varies with time period, as can be seen in the coloring of Fig. 4. For example, the baseline at the beginning of the inside experiment in Fig. 4E has a worse fit than the baseline closer to the end of the experiment. To attempt to capture this change, we next examined a piecewise fit with respect to time for the baseline response. We divided the inside and outside subsets into 10 sections each, equally by number of datapoints, with each

section corresponding to approximately ten days (as the outside data set has missing data, the sections are not all the same length of time). We then fit Equations 1 and 2 to each section, and then collected the overall fits and evaluated RMSE and $R^2$.

The piecewise Equation 1 approach resulted in $R^2$ of 0.98 and 0.99, and RMSE of 0.78 and 0.89 k$\Omega$ for the inside and outside subsets respectively. The piecewise fit is better than the full-dataset approach, but some obvious issues are still visible, such

as variation in the fit in Fig. 4B with later elapsed times.

The piecewise fit for Equation 2 resulted in $R^2$ better than 0.99 for both sets, and RMSE of 0.30 and 0.54 k$\Omega$ for the inside and outside data respectively. The fit is the best quality of the options considered, with relatively minor error and a stronger fit than Equation 1. Accordingly, we believe that including sensors to monitor non-target gasses and regularly recalibrating the baseline

sensor response to capture changing environmental conditions will be helpful in deploying TGS2611-E00 and similar sensors.

As the piecewise Equation 2 provided the best fit of the baseline regression candidates examined, we used it to produce the estimated baseline TGS2611-E00 response for our methane calibration.

### 3.3 Methane response

We examined the relationship between the sensor to baseline ratio and methane levels. As the inside dataset had a wider range of methane concentrations than did the outside set, we first fit only the inside data.

The TGS2611-E00 sensor response relative to the predicted baseline given by the piecewise Equation 2 should be proportional to methane levels. Accordingly, we use Equation 3 to estimate methane levels from the sensor response. We fit Equation 3a

on the inside dataset, and then rearrange the terms to produce Equation 3b, which we evaluate for $R^2$ and RMSE. The fit on the inside dataset shows moderate performance, with $R^2$=0.46 and RMSE=0.65 ppm. As seen in Fig. 5A, the fit is noisy but captures methane trends. The largest errors occur in the low concentration range, with the model both over and underpredicting some datapoints. The low concentration range is also overrepresented in the dataset. Despite this, the model does not have notable bias in either the lower or higher concentrations.


The 1% of points with the worst fit, which we designate as outliers as highlighted in Fig. 5B, dominate the model error; neglecting these points, we find the reasonable performance of $R^2$=0.63 and RMSE=0.53 ppm. Some of the outlier data appears to have related trends, such as the group of overpredicted points in the upper left part of Fig. 5B. To better understand the





cause of these errors and to understand possible difficulties for our model, we examined these outlier points. We found three

main explanatory features of the outlier data.

First, as seen in Fig. 5C, we calculated the absolute change in methane concentration with the data immediately before and after each point, and took the larger change of the two. Some of the outliers show a large rate of change; as our system is passive and relies on natural air movement and gas diffusion, it is possible that our sensor node did not respond to changes

quickly enough to track the relatively rapidly shifting methane concentrations for these points. 40% of the outliers show a rate of change greater than 1 ppm per 10 minutes; 31% exceed 2 ppm per 10 minutes; and 6% exceeded 5 ppm per 10 minutes.

Second, as noted previously, the methane concentrations exceeded the limits of our reference analyzer on several occasions. After each such spike we removed one hour of data to allow the systems to stabilize and resume normal functioning. We had

additional gaps from occasional power failures or system downtime for data collection. 7% of the outliers fell immediately after such data gaps, as compared to the 0.34% of all points which occurred after a gap. Our calibration model tended to overpredict these points, as seen in Fig. 5D, suggesting that possibly our sensor node was still encountering a higher concentration in the sensing chamber than in the ambient air.

Finally, we found three runs of consecutive outlier points, highlighted in Fig. 5E - these periods each had a stretch of outlier points without breaks for one to several hours each. These three events showed patterns that appeared to be related, and so next we examine them in more depth. Together, these three events account for 45% of the outliers.

The three explanatory categories had some overlap - some data from one of the anomalous events might have had a high rate

of change, for example. In total, 87% of the outliers either belonged to one of the three consecutive events, fell after a data gap, had a rate of change greater than 1 ppm per 10 minutes, or some combination.



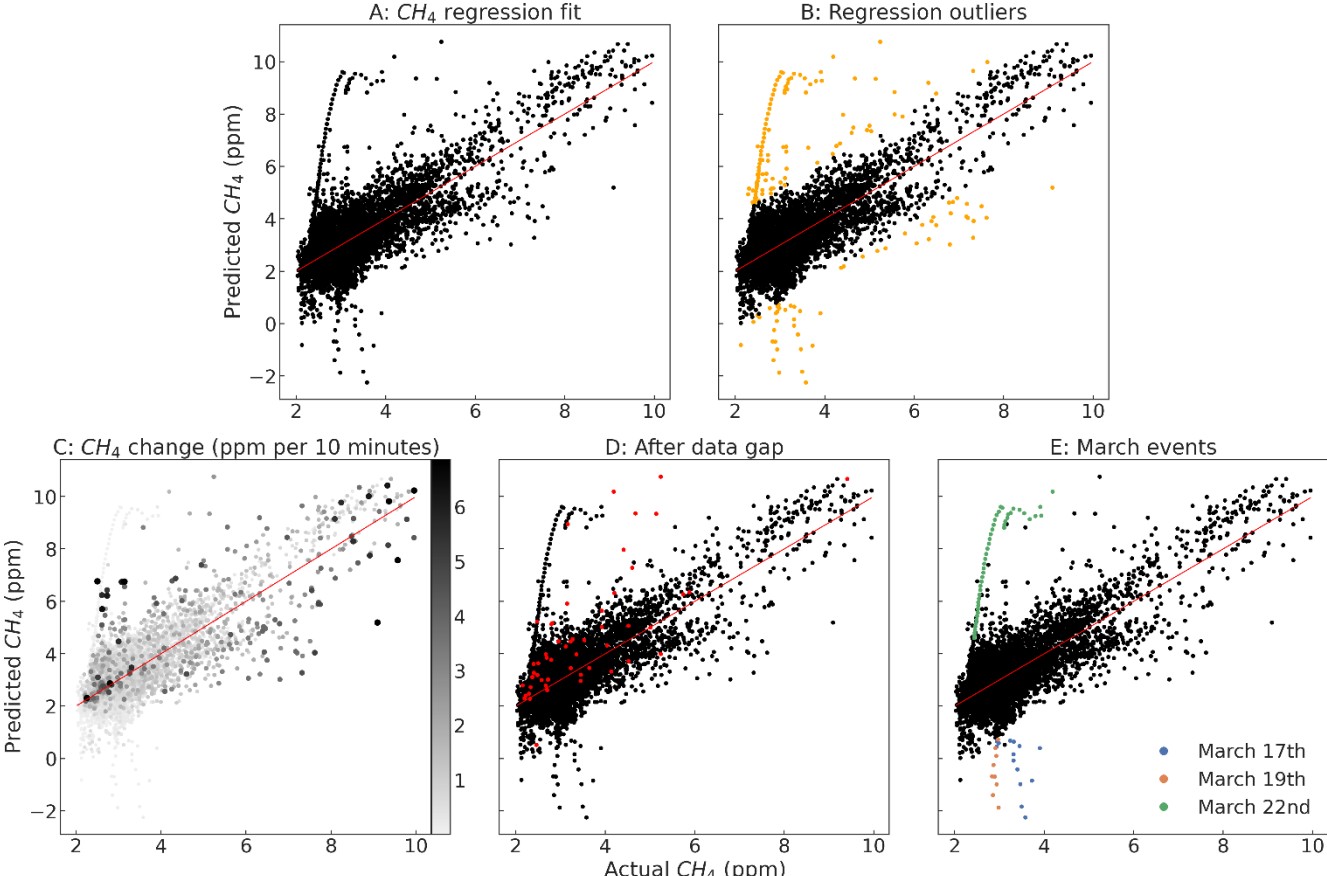

**Figure 5: Methane fit results (A), 1% outliers (B), points colored by rate of methane change (C), occurrence after a data gap (D), and the three events in March discussed in the text (E).**

Figure 6 plots the three consecutive outlier events. The first two events - on the 17th and 19th of March - show similar behaviors: in both cases the TGS2600 sensor exhibits a sharp and strong response to something other than methane or humidity, without the TGS2611-E00 sensor responding. As the TGS2611-E00 has a filter to remove certain interfering gasses and the TGS2600 does not, we speculate that this response is likely the result of an unknown, non-target gas pulse. This strong signal causes an error in the baseline regression, leading to an erroneous methane prediction.


The third event, on the 22nd of March, shows different behavior. Typically, the MOx sensors respond inversely to both humidity levels and methane; in the depicted event, both sensors show an unexpected response through the day that is both stronger in magnitude and mostly in the opposite direction from the expected humidity response. The sensor response is also much larger than we expected from the depicted methane fluctuations. We speculate, again, that the sensors were responding

to some unknown gas, with a gradual release through the day and slow dissipation overnight.


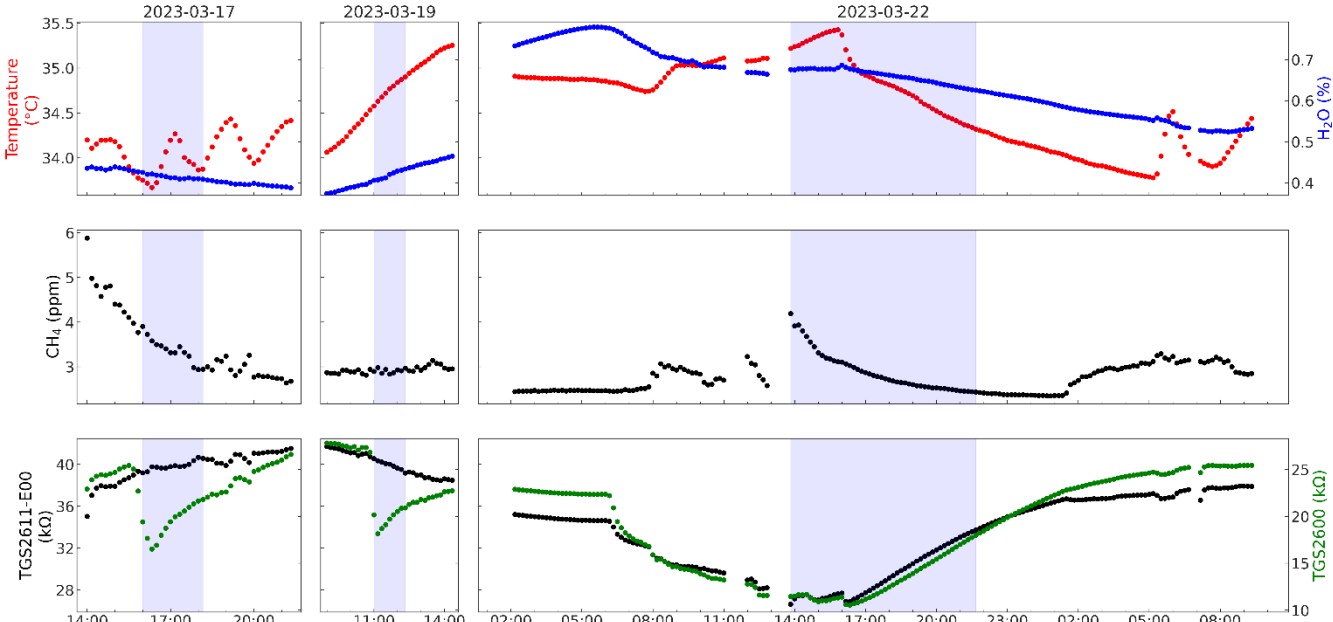

**Figure 6. Time series plots of the three anomalous events. The light purple highlights represent consecutive periods of outlier points, as described in the main text.**

Finally, we attempted the same approach on the outside dataset, fitting the baseline with the piecewise Equation 2 and then predicting methane with Equation 3b. As noted above, the baseline regression produced a strong fit, but the fit for methane did not capture any trend, and failed to perform better than simply predicting the dataset mean. 97% of the outside data had a methane concentration between 1.98 and 2.5 ppm, as compared with the 45% of the inside data in the same range; as the RMSE for our inside methane model was greater than 0.5 ppm, the outside data appears to cover too small of a range to accurately fit.

### 3.4 Methane regression sensitivity to baseline fit

We fit the TGS2611-E00 baseline response with Equations 1 and 2, both over the full dataset and piecewise over time intervals - as a reminder, Equation 2 included the TGS2600 sensor response as a term, while Equation 1 did not. We derived the sensor methane response using the piecewise Equation 2 baseline fit.

To examine whether this baseline fit is critical for sensor performance, we performed the methane calibration routine we described for the four possible baseline fits on the inside dataset, with results shown in Fig. 7. All of the regressions capture some trend in the methane response, but only the Equation 2 piecewise fit (the approach we examined in detail previously) has $R^2$ better than simply predicting the dataset mean for each point; the other baseline equations lead to poor methane predictions in the low range, causing a poor overall fit. Accordingly, it appears that tracking the changes in sensor baseline response to environmental conditions will be critical to successful field deployment.





**Figure 7: Sensor calibration results using the different baseline regression approaches.**

## 4 Discussion

### 4.1 Sensor baseline determination and methane fit

We found that we could closely fit the baseline TGS2611-E00 response to environmental conditions at background methane
levels. Water vapor, temperature, and sensor runtime appear to largely determine the sensor baseline ($R^2 > 0.9$). The baseline





fit performs better if approached piecewise with respect to time. The piecewise approach caused discontinuities between baseline predictions at the edge of adjacent calibration periods; a more sophisticated approach might interpolate regression parameters between calibration points, but we accepted the simple method as sufficient for this paper.

The sensor deviation from baseline showed a clear trend with methane levels, without notable bias, but with substantial uncertainty. The model appears adequate to capture substantial shifts in methane concentration, but appears too noisy to monitor small changes. Our model can likely distinguish 2 from 10 ppm, but appears unable to distinguish, for example, 2 from 3 ppm. We speculate that some of the model's error is caused by changes in non-targeted ambient gas concentrations; previous studies have also noted this effect (Shah et al., 2023). We also believe that the passive nature of our sensor node may

have caused a slower methane response than the active sampling reference analyzer; adding a pump to our sensor node would likely improve temporal resolution and reduce this lag or mismatch. Our sensor design showed excellent power supply stability and had good electrical resolution; accordingly, we think it is unlikely that the electronics contributed substantially to model error.

Our fit for methane as a function of sensor response showed moderate performance in the 2 to 10 ppm range we examined. Our calibration equation captured methane trends (neglecting outliers, $R^2=0.63$), but the previously mentioned difficulties and resulting outlier points led to compromised performance (with outliers, $R^2=0.46$). As the baseline fit was piecewise with respect to time, attaining this level of performance may require frequent recalibration of the sensor response to environmental factors.

We were similarly able to closely predict the baseline sensor response for our outside dataset, which predominantly had data close to background levels (2 ppm methane), but were not able to capture methane trends. We believe that this is likely a limitation of the sensors themselves, and that concentrations in the background to 2.5 ppm range, as encountered in our urban site, will be a poor application for this technology.

Various authors have claimed that TGS-series sensors require moderate relative humidity to operate consistently, with a threshold given of approximately 40% (Eugster and Kling, 2012; van den Bossche et al., 2017; Riddick et al., 2022). Our inside dataset, with which we found moderate success in fitting sensor response to methane levels, had a maximum relative humidity of 18% with a mean of 10%; the low humidity levels were due to the large, heated but not humidified study location in winter. As discussed previously, although humidity influences sensors' performance, it appears that unmonitored parameters

other than humidity were also responsible for sensing difficulties; accordingly, it is not apparent that low humidity poses a fundamental problem for the application of TGS2611-E00 or similar sensors.



## 4.2 The importance of time

Some previous studies have found that different algorithms or parameters were required at different time periods, or that models developed in the lab demonstrated compromised performance in field experiments (Collier-Oxandale et al., 2018; Riddick et al., 2020; Shah et al., 2023). Some time-related factor, then, may be important for sensor response. Time was a significant parameter in our fit for sensor baseline response in two ways: first, including a parameter for elapsed sensor lifetime improved the model quality; and second, fitting the baseline piecewise over shorter timespans led to better performance.

To examine the nature of the piecewise fit, we fit Equation 2 over the full dataset (both outside and inside) split into 20 sections. The time chunks for the piecewise fit are not necessarily of equal temporal length due to gaps in the data, but instead each contain the same number of data points. As seen in Fig. 8, the best-fit regression coefficients change in a non-monotonic manner, although with some apparent visible patterns.

We speculate that the change in best fit parameters for the sensor baseline over time is due to changing environmental parameters that our setup was unable to measure, such as interfering gasses or ambient air makeup; sensor aging may play a role as well, but as the coefficients change in a non-monotonic manner aging may not be a primary cause.

We suggest that previous difficulties implementing these sensors may have resulted from similar issues. Determining whether other ambient gasses, sensor aging, or other factors are responsible will require further investigation. We believe this underlying issue is the next problem to address for the practical use of these sensors for methane monitoring, and that possibly deployment in more extensive sensor arrays targeting different gasses will improve methane monitoring results.



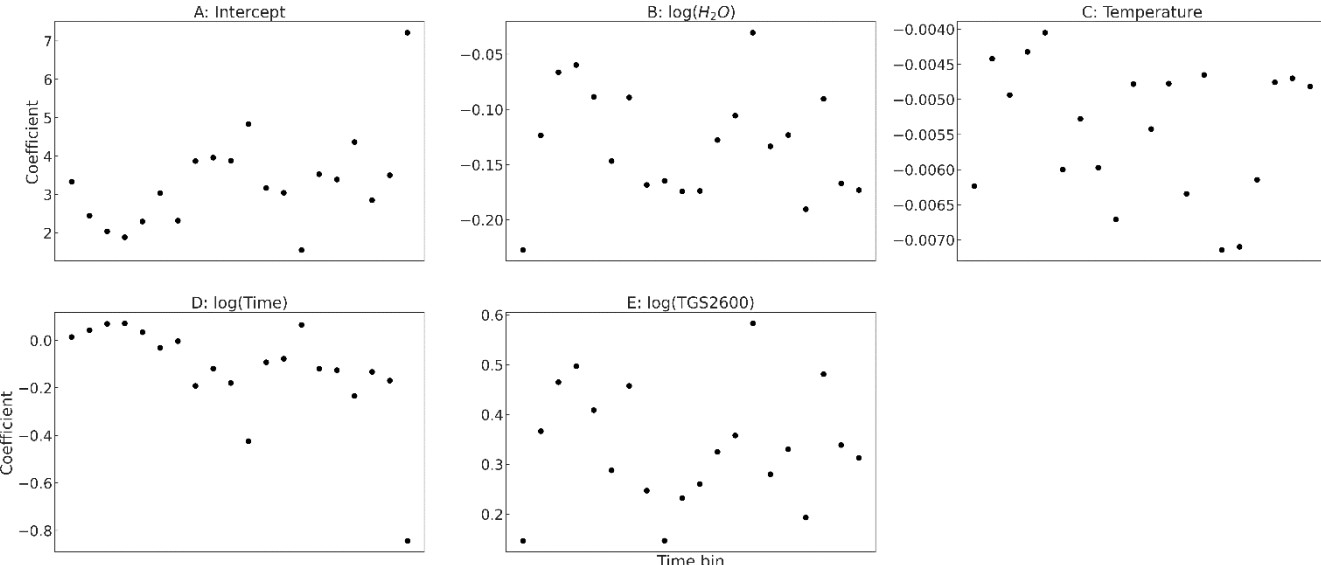

**Figure 8. Best-fit regression coefficients for Equation 2, fit across the combined dataset. As described in the text, the Time axis is not to scale. A is the intercept; B-D are environmental parameters and time, and E is the non-methane responding MOx sensor.**

**4.3 TGS2600 performance for methane sensing**

Some previous studies have used TGS2600 to monitor methane levels. We repeated our analysis, exchanging TGS2600 and TGS2611-E00 in Equations 1 and 2 for the background fit, and then again attempting to fit methane response as a function of

the TGS2600 to background fit ratio.

We found that we could closely fit the sensor baseline response, with $R^2$ of 0.98 and RMSE of 0.73 kΩ for the modified piecewise Equation 2, closely comparable to the values of 0.99 and 0.71 kΩ for TGS2611-E00. For the modified piecewise Equation 1, we found $R^2$ of 0.91 and RMSE of 1.53 kΩ. However, neither possible fit (with the Equation 1 or Equation 2

baselines) for methane had $R^2$ better than simply predicting the mean, nor RMSE better than 4.5 ppm. This result is consistent with our previous work showing little to no response from TGS2600 in the 0-10 ppm concentration range (Furuta et al., 2022). Accordingly, we believe that TGS2600 does not have promise for methane sensing in the near-background concentration range, and that previous studies using this sensor may have encountered algorithmic overfitting, interfering co-emitted gasses, or other unknown environmental effects.


The inability of TGS2600 to sense low concentrations of methane may be beneficial if it is included in a sensor array with TGS2611-E00 or a similar sensor; possibly such a sensor array could reduce any effect of non-target gas species, and give greater insight into gas compositions than possible with a single sensor.





## 5 Conclusions

Our sensor node demonstrates the feasibility of a low-cost, high-performance implementation of the TGS2611-E00 or similar MOx sensors. Our system design provided stable operating conditions for the MOx sensors for over half a year of field tests, and could be powered by an inexpensive battery and solar panel in a stand-alone deployment. Our unit had a parts cost of under \$200; future revisions could easily reduce costs further by implementing the datalogger and telemetry from components rather than using off-the-shelf microcontroller modules.


We found TGS2611-E00 to respond to methane, consistent with our previous work. Contrary to some previous studies but consistent with our previous work we did not find TGS2600 to respond to methane in the studied 2 to 10 ppm range. We suggest that work finding TGS2600 to respond to methane in a low concentration range should consider possible co-emitted gasses, algorithmic overfitting, or other experimental factors.


Our results suggest that similar sensor networks are worth investigating for applications with methane concentrations of interest above 10 ppm, such as around landfills, animal agriculture and manure processing, wastewater treatment, fossil fuel infrastructure, or other near-source settings. Our sensor response correlates with methane levels with moderate accuracy in the lower 2 to 10 ppm range, but caution is necessary to account for environmental factors. Our system was unable to capture
methane trends in the 2 to 3 ppm range in our outdoor test.

In addition to MOx sensors' well-known sensitivity to humidity levels, we found that sensor performance varied over time, possibly in response to changing ambient gas compositions, sensor aging, or other unmeasured environmental changes. We believe that this sensitivity will limit the ability of these or similar sensors to accurately monitor low methane concentrations
in many real-world settings without further work to detect and correct for other gasses or environmental factors. We believe that this behavior may ultimately determine the lower limit for practical deployment of these sensors as standalone units.

Our results could possibly be improved by filtering or monitoring and accounting for interfering gasses, and possibly by controlling water vapor levels and temperature in the sensing chamber. Converting our system to active sampling with the
addition of a pump could also help reduce the system's lag time, and might allow the unit to capture sharper methane peaks. Even with these improvements, it is likely that these MOx sensors will be better suited to methane concentrations above the near-background range.

## Appendix A: Sensor node electronic design

Our sensor node consists of two circuit boards: a main controller board and a sensor board. The two boards are connected via
a cable. The main board is responsible for telemetry, data storage, and system control; the sensor board allows for up to two



Figaro MOx sensors, a relative humidity and temperature sensor, and power regulation for the Figaro sensors. The full device is pictured in Fig. 1A of the main text.

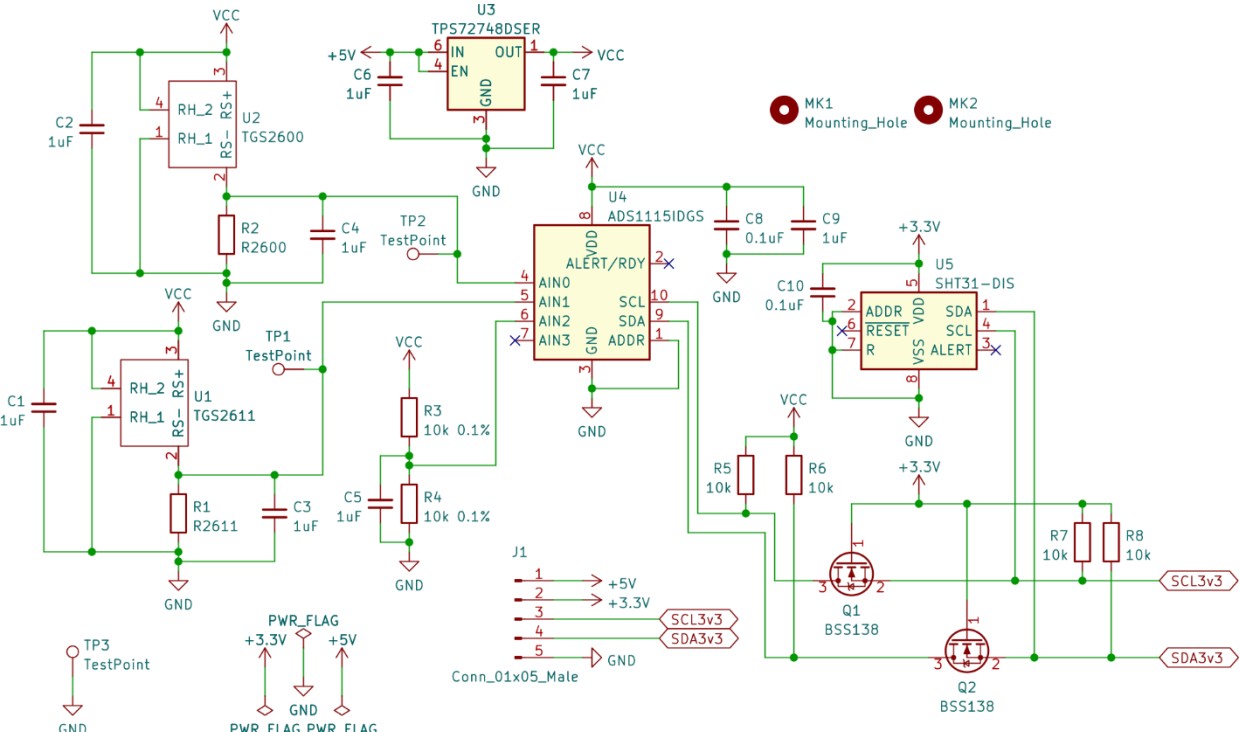

**Figure A1: Schematic for the sensor circuit board.**

Figure A1 shows the sensor circuit board. Digital communication between the controller and sensor boards is provided via I$^2$C with a high voltage of 3.3V. The Figaro MOx sensors are specified with a 5±0.2V supply (Figaro USA, Inc., 2013; Figaro USA, Inc., 2022). We provide a 5V supply from the controller board; however, as power supply stability is critical to accurate sensor readings, we further regulate the supply with component U3, a precision voltage regulator with a fixed 4.8V output. As discussed in the main text, this arrangement proved stable over the course of the experiment.


The MOx sensors U1 and U2 are implemented in voltage divider configurations with R1 and R2, which were chosen to approximately match the expected sensor resistances, as discussed in the main text. The output voltages from the dividers are digitized by U4, a 16-bit ADC. We also digitize the 4.8V supply through R3 and R4, allowing us to correct for any drift and to evaluate the regulator performance. U4 contains an internal precision voltage reference. The ADS1115 has a resolution of

188 μV, corresponding to a resolution of 12 Ω against the 75 kΩ reference resistor and 2.3 Ω against the 15 kΩ reference resistor when the sensor resistances equal the references.



We sense temperature and relative humidity using U5, an integrated, digital-output chip. U4 and U5 communicate with the controller board via I²C; as U4 is operating at 4.8V, level shifting is required for communication with the 3.3V controller

device. We provide level shifting via Q1, Q2, and the associated pull-up resistors R7 and R8.

All capacitors on the board are provided for power supply bypassing to reduce noise and instability, and all were X7R dialectic ceramic capacitors.

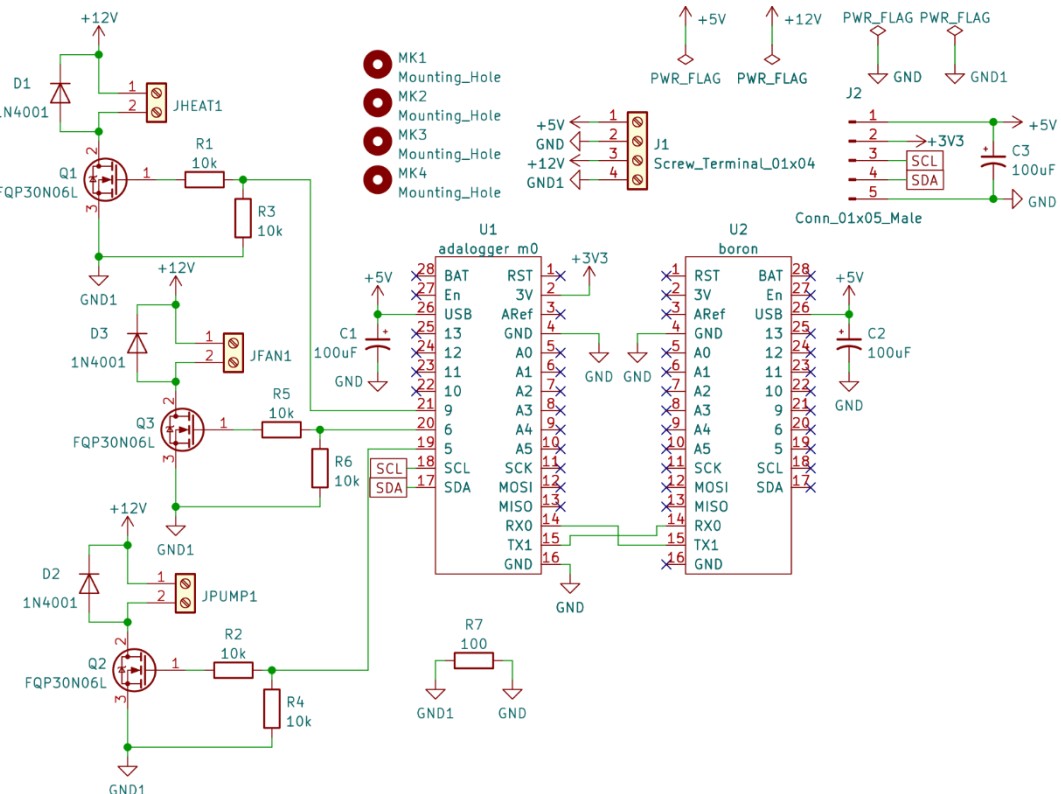

**Figure A2: Schematic for the main board.**

Figure A2 provides the schematic for the main board. U1 is a complete microcontroller module, including a processor, 3.3V power regulator, SD card socket for data storage, and supporting circuitry. We use U1 as the main controller for the whole system, as well as for data storage. U2 is a microcontroller module with a cellular modem; we only use this board for telemetry and as a networked clock. We only used the telemetry data to remotely check that the system was operating correctly, and our

analysis used the locally stored, five-second scale data.

Q1 through Q3 and their supporting components are provided to optionally drive a fan, pump, heater, or other similar devices; we did not use these components for the current experiment, and we left them unpopulated on the circuit board. As with the



sensor board, we placed capacitors C1 through C3 to bypass the power supply for lower noise and greater stability; all three

were aluminum electrolytic capacitors.

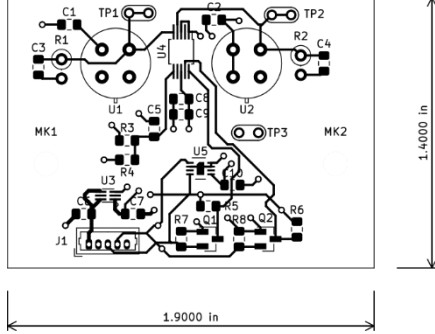

**Figure A3. Circuit layout for the sensor board.**

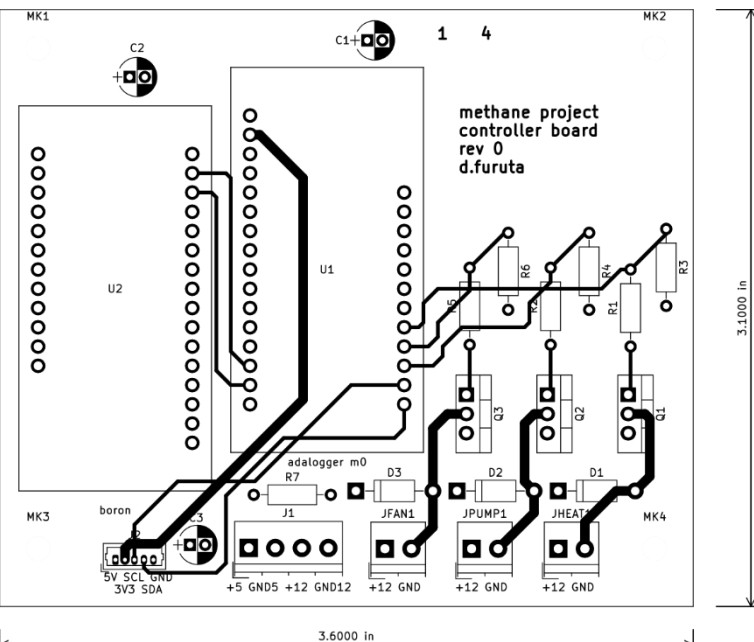

**Figure A4. Controller board layout.**

Figures A3 and A4 show the board layouts. Both boards are four-layer stacks. For clarity, the inner two layers containing ground and power planes have been omitted from the figures. We are aware of one mistake on the board layout: J1 on the sensor board should be reversed. For our prototype unit we simply reversed the cable connection to this part. As mentioned previously, the fan, pump, and heat components on the main board were not used or populated for this experiment, but are provided in the layout for future development. We hand-soldered and assembled the boards for this experiment.



## Appendix B: Baseline regression equation selection

As discussed in the main text, we examined different possible regressions to fit the TGS2611-E00 response to baseline environmental conditions other than methane. We fit the regressions on the datasets filtered to only include points with methane concentrations less than 2.3 ppm, which we assumed would be too small of an enhancement over the 2 ppm background for the sensor to respond. For the inside dataset, 3717 of the 15657 points were below the 2.3 ppm threshold, and for the outside dataset 13030 of 14330 points were included.

As explained in more detail in the main text, we have four possible factors for the TGS2611-E00 baseline regression: water vapor concentration, temperature, elapsed sensor run time, and TGS2600 response. We also examined factors transformed by the natural log; we adjusted the log-transformed time variable by a negligible amount to account for zero values. We fit all regressions possible from Equation B1 on the inside data and outside data, and both sets together.

$$y = \alpha + \beta_1 x_1 + \beta_2 x_2 + \beta_3 x_3 + \beta_4 x_4 \tag{B1}$$

Where $y \in \{TGS2611, \log(TGS2611)\}$, $x_1 \in \{H_2O, \log(H_2O)\}$, $x_2 \in \{T, \log(T)\}$,
$x_3 \in \{time, \log(time + 0.0001)\}$, $x_4 \in \{TGS2600, \log(TGS2600)\}$

We evaluated each regression for $R^2$ and RMSE. Regressions using the log-transformed TGS2611-E00 response as the target were transformed back to linear space prior to evaluating performance. We show the performance of the three best regressions on each dataset without TGS2600 and the three best with TGS2600 (equivalent to $\beta_4$ in Equation B1 taking a zero or non-zero value, respectively) for each dataset in Table B1, chosen by lowest RMSE. The regressions are shown in shorthand; for example, $log(TGS2611) \sim log(H_2O) + T + log(time)$ is shorthand for the regression equation $log(TGS2611) = \alpha + \beta_1 log(H_2O) + \beta_2 T + \beta_3 log(time + 0.0001)$.

**Table B1. The three best performing baseline regressions for each datasheet with or without TGS2600 as a term, sorted within each category by RMSE.**

| Dataset | Regression | $R^2$ | RMSE (ppm) |
|---------|------------|-------|------------|
| Outside | $log(TGS2611) \sim log(H_2O) + T + log(time)$ | 0.97 | 1.46 |
|         | $log(TGS2611) \sim log(H_2O) + \log(T) + log(time)$ | 0.97 | 1.52 |
|         | $TGS2611 \sim log(H_2O) + \log(T) + log(time)$ | 0.97 | 1.53 |
| Inside  | $log(TGS2611) \sim log(H_2O) + \log(T) + log(time)$ | 0.91 | 1.52 |
|         | $log(TGS2611) \sim log(H_2O) + T + log(time)$ | 0.91 | 1.54 |
|         | $TGS2611 \sim log(H_2O) + \log(T) + \log(time)$ | 0.90 | 1.63 |
| Both    | $log(TGS2611) \sim log(H_2O) + T + time$ | 0.90 | 2.50 |
|         | $TGS2611 \sim log(H_2O) + T + time$ | 0.90 | 2.61 |
|         | $TGS2611 \sim log(H_2O) + \log(T) + time$ | 0.90 | 2.71 |





| | | | |
|---|---|---|---|
| Outside | $log(TGS2611)\sim log(H_2O) + T + log(time) + \log(TGS2600)$ | 0.99 | 0.71 |
| | $TGS2611\sim log(H_2O) + \log(T) + log(time) + TGS2600$ | 0.99 | 0.75 |
| | $log(TGS2611)\sim log(H_2O) + \log(T) + log(time) + \log(TGS2600)$ | 0.99 | 0.76 |
| Inside | $log(TGS2611)\sim log(H_2O) + \log(T) + log(time) + \log(TGS2600)$ | 0.98 | 0.71 |
| | $log(TGS2611)\sim log(H_2O) + T + log(time) + \log(TGS2600)$ | 0.98 | 0.71 |
| | $log(TGS2611)\sim log(H_2O) + \log(T) + time + \log(TGS2600)$ | 0.98 | 0.76 |
| Both | $log(TGS2611)\sim log(H_2O) + T + log(time) + \log(TGS2600)$ | 0.98 | 1.21 |
| | $TGS2611\sim log(H_2O) + \log(T) + log(time) + TGS2600$ | 0.98 | 1.27 |
| | $TGS2611\sim log(H_2O) + T + log(time) + TGS2600$ | 0.98 | 1.27 |

As seen in Table B1, several equations occur repeatedly; in particular, the top three equations without TGS2600 are the same
for the inside and outside datasets, in different orders but with similar RMSE and $R^2$ values. As
$log(TGS2611)\sim log(H_2O) + T + log(time)$ performs well for both the inside and outside datasets, and as it performs best
across the datasets with the added $\log(TGS2600)$ term, we chose this regression to use as our Equation 1 in the main text, and
the version with added $\log(TGS2600)$ to use as Equation 2.

**Appendix C: Diurnal patterns in methane levels**

We observed diurnal patterns in methane levels in some portions of the outside dataset, as seen in Fig. C1C1, with increases
in methane levels at night and lower levels in the day. The time period depicted was a rainy week; diurnal patterns were smaller
or absent in dry periods.

Methane levels at our inside site also fluctuated, as seen in Fig. C1C2. These fluctuations appeared to be primarily the result
of interactions with the anaerobic digester, with pulses occurring when the digester was opened for feeding or gas removal.
The digester also likely released methane sporadically when internal pressure was released through its pressure relief system,
which was simply a tube submerged in several inches of water.



**Figure C1. Examples of short periods of the outside (A1-D1) and inside (A2-D2) experiments.**

**Code and data availability**

Code and data are available at https://hdl.handle.net/11299/258238.



**Author contributions**

DF and JL designed the experiment, with input from all authors. DF designed and fabricated the sensor node and performed the experiments. All authors contributed to data analysis and model building. DF prepared the manuscript with contributions
from all authors.

**Competing interests**

Albert A. Presto is a member of the editorial board of AMT. The authors have no other competing interests to declare.

**Disclaimer**

This publication was developed in part under Assistance Agreement No. 84062701-0 awarded by the U.S. Environmental
Protection Agency to Jiayu Li. It has not been formally reviewed by EPA. The views expressed in this document are solely those of the authors and do not necessarily reflect those of the Agency. EPA does not endorse any products or commercial services mentioned in this publication.

**Acknowledgements**

This research has been supported by funding from the U.S. Environmental Protection Agency's Understanding and Control of
Municipal Solid Waste Landfill Air Emissions program.
This research has been supported by the National Energy Technology Laboratory (grant no. S000663-USDOE).

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
