# Peer review of "Design and evaluation of a low-cost sensor node for nearbackground methane measurement"

_Atmospheric Measurement Techniques, 2023_

## Referee Comment (RC1)

Dear Editor,

This manuscript tests a low-cost sensor node containing a Figaro TGS 2600 and a Figaro TGS 2611-E00 metal oxide sensor, alongside a high-precision reference instrument. The sensor was tested both outside and inside. The outside dataset could not be used due to a lack of methane enhancements, despite efforts to artificially enhance the background. The inside dataset could however be used. The correlation of both sensors to various environmental conditions and methane concentration were presented, using the TGS 2611-E00 as the main methane sensor. This work took the approach of deriving a baseline resistance corresponding to methane concentration below 2.3 ppm, to incorporate non-methane environmental effects, which included a time component. A second baseline was also devised which included TGS 2600 measurements, as this sensor was assumed not to be methane-sensitive. The ratio between measured resistance and baseline resistance compared to methane concentration was poor for both approaches. Instead, the baseline was split into 10 pieces for which individual baseline coefficients were derived. This approach was slightly better. A sensitivity analysis was conducted on piecewise baseline variability components over time. In another analysis, the TGS 2600 was also treated as the main methane sensor, but this did not yield results better than treating the TGS 2611-E00 as the main sensor. In any case, any analysis on methane concentration is slightly dubious as most methane concentration measurements were obtained in background conditions. The greater value of this work is in baseline resistance modelling. This paper is a welcome addition the growing body of literature on this subject. I was particularly impressed by the meticulous nature of this work. The standard of English and the presentation is very good. I suggest the following editions to the work, in addition to the two paragraphs below, where I address two key issues.

Using the TGS 2600 resistance measurements to derive a baseline fundamentally requires for TGS 2600 methane sensitivity to be negligible. The theory is that this (assumed) non-methane sensor can be used to characterise non-methane effects. Yet, if methane effects are not negligible, then methane sensitivity may be reduced in the resistance to baseline ratio, thus being counterproductive. Based on visual inspection of figure 7, there is not much improvement when comparing an eq 1 (no TGS 2600) baseline to an eq 2 (with TGS 2600) baseline versus methane concentration. The authors do not provide $R^2$ values for each fit and only provide values for their favoured fit (eq 2 piecewise). Unless the authors can prove that the eq 1 baseline approach is significantly better than the eq 2 baseline approach, I do not think using TGS 2600 measurements within a baseline is necessarily better. The authors' justification for using eq 2 instead of eq 1 is that it results in a better baseline fit (*i.e.* modelled baseline versus measured baseline). This is hardly surprising as this simply shows that two similar TGS units behave in a similar way. But the authors must also be sure that there is no TGS 2600 methane effect.

To test the TGS 2600 methane effect, the authors do apply a baseline correction to the TGS 2600 data (in the same way for the TGS 2611-E00 data), observing poor methane correlation. Yet, the authors fail to provide an $R^2$ value of the methane fit for comparison. Furthermore, most TGS 2611-E00 baseline attempts also failed, when compared to methane concentration. I think the 10 ppm limit and the number of data points may have been insufficient to reveal sufficient correlation for this sensor. I therefore question the conclusion from this manuscript to dismiss the utility of the TGS 2600 from further research. The authors must clearly state these caveats and limitations when discussing their views and outlook on the use of the TGS 2600. This work is not sufficient to conclude that the TGS 2600 is not sensitive to methane below 10 ppm.

**General comments**

Abstract
- This is concise and easy to follow. However, this section is written slightly vaguely. The abstract should include more specific details on the work presented in the manuscript and not be afraid to use a bit of technical language. For example, a crucial advancement in this work is the development of a baseline resistance model. This should definitely be in the abstract. The authors could also add some more details on the enclosure (such as size), if appropriate.
- I also think that the duration of testing time should be included here.
- I am interested by the decision not to specify that a Figaro TGS is being used either here or in the title. I do not insist that the authors do so. However, I would appreciate some rationale please, as I do not understand the value of this.

Introduction
- This section provides a very good summary of recent research on the TGS in methane concentration measurements, with good referencing. However, this first paragraph could provide a better overall motivation for the necessity of this work in the context of increasing methane emissions, with some citations.
- The way in which the narrative of this section is conveyed is sometimes confusing as it is not clear whether it is written in the context of methane or more generally. Based on the first sentence, this is a paper on methane measurements. It should therefore be emphasised throughout this section that points are being made in the context of methane sensors and not more generally.

Methods
- Overall, this section is well-constructed and covers the methods very well. The calibration section was particularly well-written and easy to follow.
- Regarding the timestamps, was the lag time of the TGS enclosure ever tested? The 30 s lag time of the LI7810 was assumed to be negligible, which is fine. However, the lag time of the TGS logger is a bigger issue. If a spike of methane was emitted just beneath the TGS enclosure, I wonder how long it would take TGS resistance to peak in response. It would not be short (a few seconds), as air is carried to the sensors by diffusion. I do not think 10-minute averaging will overcome this problem, unless TGS peak time is less than 1 minute. The averaging is however still necessary to account for the nature of diffusion, which is not instantaneous like a LI7810. The overall solution is both a lag-time correction in combination with averaging to smooth any emission spikes.

Results
- The investigation of various causes of baseline disparity is a welcome inclusion in the research on this topic. The correlation plots and the baseline analysis are both good. However, the work on a methane response using different baseline approaches lacks detail.
- In addition, this section must include an analysis of methane correlation excluding periods used to produce a baseline fit. Otherwise, it is mostly an analysis of the ability of the baseline to predict the baseline as the vast majority of datapoints were below 2.3 ppm, *i.e.* the baseline threshold.

Discussion
- The sensitivity analysis of the time component on the baseline is a very useful and robust way of identifying the nature of temporal variability.
- Testing the ability to derive a TGS 2600 baseline for methane correlation is a very good idea. However, the caveats must be discussed more clearly here. Furthermore, a plot of methane correlation would be welcome. It is not clear how much worse the TGS 2600 is than the TGS 2611-E00, when detecting methane.

Conclusion
- This is a nice overview of the work, but lacks a few key values and outcomes from the work, that could be included.

**Specific comments**

Line 11: We deployed the prototype sensor alongside a reference methane analyzer in two sites - one outdoors, one indoors - for several months each of data collection across a range of environmental conditions and methane levels.
- This sentence is difficult to follow.
- Also, please specify which types of environmental conditions. Variations in temperature?

Line 13: calibration models
- Please add a few very brief details on what sort of calibration models. What is the crucial basis of the calibration approach? Were linear models used? Was machine modelling used?

Line 14: background monitoring and enhancement detection
- Monitoring and detection of what? Presumably this is methane concentration.

Line 14: performance
- What does "performance" mean? Does this mean accuracy compared to the reference instrument? Please clarify.

Line 14: 2 to 10 ppm range
- Please specify that this refers to methane concentration.

Line 17: these and similar inexpensive MOx sensors
- Of which sensors? It is not specified which sensors have been used. The authors cannot refer to "these" sensors and compare them to similar sensors, if the sensor has not yet been introduced.

Line 17: near-background methane monitoring
- What does this mean? Perhaps state that this refers to up to 1 ppm (or whatever) concentration enhancement.

Line 19: scarcity of high-resolution ground-level data
- Please provide some references for this statement.

Line 21: variety of sensor mechanisms
- Please specify that this refers to methane sensors. It is better to write this section in the context of methane detection, rather than general low-cost sensor use.

Line 29: sensor array or "e-nose" configurations
- Please briefly explain how using a sensor array can overcome issues of selectivity.

Line 38: when calibrated in a laboratory setting
- Please specific that it was both calibrated and tested in a laboratory setting. This 1.7 ppm value refers to an indoor laboratory test.

Line 40: Cho et al. (2022) find
- Please write this in the past tense.

Line 41: above 100 ppm concentrations
- In the laboratory or in the field?

Line 41: 2 to 100 ppm range
- Again where? In the laboratory or at a glacier?

Line 42: but found that quantitative emission estimates had poor accuracy
- I don't understand the relevance of this. This is a manuscript on methane concentration measurements and not flux measurements.

Line 48: positive results
- Please clarify, positive results in what? In methane concentration measurements? It is previously stated that this sensor can be used to measure many gases, so this must be clear.

Line 48: Eugster and Kling (2012) report a
- Write this in the past tense.

Line 49: Other papers find
- Write this in the past tense.

Line 83: We burned in the sensors and regulator for a week prior to data collection.
- Why

Line 91: Our system performed well in both sections of the experiment, with 95% of all data showing a sensor supply voltage within ±0.25 mV of the mean and 99.99% of all data showing a supply voltage within ±0.80 mV of the mean across the full dataset.
- This is a very good demonstration of good practice and an excellent result. It is impressive.

Line 119: Our first site was an urban yard in Minneapolis, USA
- Perhaps provide the coordinates, if possible.

Line 125: The background methane concentration at our research site
- How is this known? Is this based on the LI7810? Please specify.

Line 127: 2.5% methane gas cylinder
- What was the rest of the cylinder filled with Synthetic air? Natural air? Argon? Nitrogen?

Line 129: These releases produced a maximum 10-minute averaged methane value of 5.8 ppm, with most of the releases producing methane concentrations between 3 and 4.5 ppm.
- Again, please specify how this is measured. Is this based on the reference gas analyser?

Line 133: Our second site was indoors in the Biosystems Engineering building at the University of Minnesota, Twin Cities campus.

- This is a little confusing. Quite simply, one test was conducted was outside and one test was inside the building. However, it reads as if there were two totally different sites in different locations, which adds to the complexity. "Our second site" implies that there were two totally separate testing locations.
- In reality, there was only one site, not a first and second site.

Line 136: We also expected possible emissions of methane and other gasses from surrounding labs

- This could be a crucial point. The lab was filled with potential interfering gas sources, which could influence the TGS response. I think the authors may elaborate here on potential gases that could be present or, at least, provide some more specific details on the activities taking place in the vicinity.

Line 153: we removed two hours of data after each reboot

- Can the authors confirm that 2 hours was sufficient for the sensor to stabilise after power loss? I suggest a plot showing this.

Line 156: dataset averaged to a 10-minute timescale

- Was it a 10-minute running average or were 10-minute averages taken next to each other. Please clarify this both here and throughout the manuscript.

Line 172: We chose to include sensor run time to incorporate any effects of sensor aging;

- This is a very good idea. Is this the absolute time or the time of the sensor being switched on to account for data gaps?

Line 177: Relative humidity is dependent on water vapor concentrations and temperature

- And also, pressure.

Line 179: we decided a priori to include water vapor concentrations and not relative humidity as a possible term in our analysis.

- I totally agree with this rationale. Perhaps cite some other research that preferred to use water concentration instead of relative humidity.
- Another important detail is missing here. Is this water concentration from the LI7810 or is it derived from the SHT? If it is derived, the authors must state how it derived.

Line 193: The TGS2611-E00 sensor response is expected to deviate from the predicted baseline response as a result of methane levels as shown in Equation 3a.

- Why was this equation chosen? It is based on any previous work? I have never seen a purely linear fit applied to a resistance ratio. This is no criticism, but I am interested to know where it comes from.

Line 212: We observed a diurnal cycle in methane levels

- How? Using the reference gas analyser?

Line 213: soil processes

- What does this mean?

Line 240: humidity

- What does this mean? Is this water concentration? The term "humidity" is used regularly throughout the manuscript. However, I do not think this is wise as it could mean either water concentration or relative humidity. I suggest that the authors replace all "humidity" terms in this manuscript with a more precise term (either water concentration, relative humidity or something else) to avoid ambiguity.

Line 248: relative humidity

- I'm not too sure in the value of including relative humidity here in this list and in the previous analysis. As the authors themselves state, relative humidity and water mole fraction are related. So, it is hardly surprising that both influence TGS resistance.

Line 252: by including TGS2600 in the baseline response, we can possibly remove influence from nontarget gasses and other unexpected factors.

- The authors also remove the baseline from the TGS2600 dataset? This should be mentioned in this section somewhere.

Line 260: 1.46, 1.56, and 2.81 kΩ

- These values should include a plus-minus sign as the square root has a negative and positive solution. The same for RMSE below.

Line 293: The fit is the best quality

- Which fit?

Line 306: with R2=0.46 and RMSE=0.65 ppm

- The authors should also provide RMSE and $R^2$ values for periods not used to derive a baseline (*i.e.* above 2.3 ppm). This would help to evaluate the capability of the approach when not applied to the same data used to derive a baseline fit.

Line 317: we calculated the absolute change in methane concentration with the data immediately before and after each point

- What is the time distance between each point? Are they 10-minute running averages of averages next to each other?

Line 338: occurrence after a data gap (D)

- What are the red dots? I think this figure could be improved with the different colours given as a legend in each subplot. The figure caption is not clear as all of the points are plotted.

Line 371: Figure 7: Sensor calibration results using the different baseline regression approaches.

- Please change the axis from "actual CH4" to "LI7810 CH4" or similar, as "actual" doesn't mean much.

Line 439: However, neither possible fit (with the Equation 1 or Equation 2 baselines) for methane had R2 better than simply predicting the mean, nor RMSE better than 4.5 ppm.

- Unless $R^2$ values are provided, corresponding to the plots in Figure 7, it is not clear whether the TGS 2600 is truly worse than the TGS 2611-E00. The non-piecewise baseline attempt for the TGS 2611-E00 also failed and only one model worked.

Line 452: parts cost of under $200
- This is very impressive. It should be included in the methods section.
- Also, please provide the cost of telemetry data transfer in the methods section. Presumably there is a regular subscription charge of some nature.

Line 457: We suggest that work finding TGS2600 to respond to methane in a low concentration range should consider possible co-emitted gasses, algorithmic overfitting, or other experimental factors.
- I do not think that this sentence should be in the conclusion as they are not key findings from this work and are simply the opinion of the authors. We do not know for sure why the TGS2600 has worked in previous studies

Line 462: above 10 ppm
- Why above 10 ppm? This work was below 10 ppm. All available data above 10 ppm were excluded from the analysis so I do not understand why this point is being made.

Line 463: Our sensor response correlates with methane levels with moderate accuracy in the lower 2 to 10 ppm range
- This is misleading for two reasons.
- First, the authors must be precise what they mean by "moderate accuracy".
- Second, most of the analysis was performed on data below 2.5 ppm. Only a small percentage of data was anywhere near 10 ppm. So, the sensor was mostly tested at background levels and not mostly over a range up to 10 ppm.

Line 474: controlling water vapor levels and temperature in the sensing chamber
- Why would this help if the baseline model already accounts for these effects? Surely the problem is unknown environmental factors. What is the use of controlling known variables?

Line 476: methane concentrations above the near-background range
- What does this mean?

---

## Referee Comment (RC2)

Review of Furuta et al. "Design and evaluation of a low-cost sensor node for near-background methane measurement"

This manuscript designed a low-cost methane detector through combining Figaro TGS2600 and TGS2611-E00, temperature and humidity sensing and evaluated its performance range 2-10 ppm for both indoor and outdoor by a high precision methane instrument. The impressiveness of this work is the process of baseline, language expressions and meticulous data processing such as consideration of time drift, resistance conversion, voltage control and so on.

There are three good attempts on the baseline. 1. The time factor is taken into account in the baseline. 2. The author considers TGS2600 as the baseline reference for TGS2611. This is a well attempt based on the assumption that TGS2600 is insensitive to low concentrations of methane. 3. The author divided the data into 10 groups to construct baselines and correlation functions respectively. The 10 groups improved the fitting regression results based on a group of baselines.

However, the time factor and the input of TGS2600 are not significant contribute the estimating of methane. A few conclusions in the manuscript require more robust evidence. Besides, the applicability of similar devices mentioned in this MS in different environments and the drive factor leading to the difference between predicted and measured methane need to be further addressed. Overall, I think this paper is clear and well structured. I recommend it be published after addressing below comments.

**Specific comments:**

1. Line 14: 'moderate performance' could be expressed more quantitatively.

2. Line 19-20: It seems that the lack of high-resolution data and low-cost network have few causations. In addition, could you provide some stronger evidence to support the lack of high-resolution data?

3. Line 75-76: Based on the previous experiments, it is a good idea to exclude the influence of environmental factors through the response difference between the two TGS sensors, thus enabling the monitoring of methane in low concentrations (1-10ppm). However, it is well known that TGS2600 is sensitive to hydrogen, ethanol, iso-butane, carbon monoxide (CO), methane, relative humidity (RH) and temperature, while TGS2611-E00 is only sensitive to methane, hydrogen, ethanol, RH and temperature owing to its additional filter covers. (see FIGARO TGS 2611-E00&2600 product information, the above factors are ranked in order of influence.).

   And therefore, does the effect of hydrogen, ethanol, isobutane, and CO be taken into account in evaluation of the application potential of the TGS2611-E00 in low concentration situation? Especially in the indoor food experiment room. The differences in principle caused by physical between the two sensors and the response characteristics of both two resistance should be presented here.

4.  Line 115: In Fig 4B, where is the indoor sample gas inlet of $CH_4$ reference instrument? It may affect time delay.

5.  Line 119: As mentioned above, TGS2600 is also responsible for hydrogen, ethanol, isobutane, and carbon monoxide, and therefore nearby sources of relevant gas emissions or its surrounding environment are needed to be additionally stated.

6.  Line 122: Why these data are averaged at 10 mins scale rather other time scale? Is this a universal practice? or is it tested with some experiment then after 10 mins is selected? In addition, what do you means of 10 mins? Average all the data over 10 mins after then record a valid value?

7.  Line 132: In the title of Sect 2.2.1, the author reports the situation of increasing background concentration, and how many ppm the concentration is it?

8.  Line 134-135: A brief description of laboratory-generated gases with potential effects on resistance is needed.

9.  Line 173: 'time parameter'. Is the time parameter taken for the fit calculated from the start time of the per interval? In fact, within a 10-day window, environmental factors do not increase or decrease linearly but time does. Adding the time factor is a good attempt, but what is the contribution or significance of adding this factor?

10. Line 182: Is '2.3 ppm' the optimal selection after lots of tests? Or was it chosen randomly by the author? It would be an important impact factor on fitting result.

11. Line 185: I have seen lots of fitting functions in Appendix B, but can you explain why log is used rather than others? Has any other research used a similar fitting formula before? Or is the log more appropriate after statistics? Some explanation is needed here.

12. Line 212: Authors report that a diurnal cycle during rainy week is caused by soil process, please provide some solid evidence for this diurnal cycle driven by temperature.

13. Line 277: Statistical indicators are recommended to be presented on the figure, and therefore looks more intuitive. In addition, the numbers of data on each sub-figure need to be represented. The first impression is that the numbers of data in ABCD is different from EFGH.

14. Line 281-283: 'Even with the additional sensor term, the accuracy of the regressions varies with time period, as can be seen in the coloring of Fig. 4. For example, the baseline at the beginning of the inside experiment in Fig. 4E has a worse fit than the baseline closer to the end of the experiment.'.

    In Figure 2-A2, from early Apr to the end May, in this period temperature and $H_2O$ fluctuate over a wide range. While before 1 Apr, temperature and $H_2O$ changes obviously smaller, especially temperature. When the author only used only one piecewise to fit the whole study period, and the fitting result will obviously be affected by the number of samples. The former months are about 3 times than the latter months. Thus, it can be expected that the fit will be significantly worse in April and May, i.e. bring greater RMSE. And the data prefer to attribute this result to its representative rather than time.

In addition, in both Figures 4A and 4B, even if the author uses piecewise fitting in 4B, it can be clearly seen that the RMSE of the yellow points in both 4A and 4B is larger than the RMSE of the green points. Compared Fig-2A to Fig-2B, temperature and $H_2O$ in former months (Jan to Apr) also well show larger variation than latter months (after Apr) and therefore imply different possibility with author.

15. Line 293-295: I agree with baseline needs to be regression and is also a good experiment and perspectives. But the drive factor should be clarified.

16. Line 320: How about the averaged change speed of $CH_4$? Actually, I see lots of black point (high change speed points) on the red line.

17. Line 354-358: From Fig 2, the temperature and $H_2O$ in outside vary great in a day. And from Fig 3 data also Line 368 of MS show that resistance is sensitive to temperature and $H_2O$. Why not have a try on constructing a fitting baseline by interval of temperature and $H_2O$? It might get more interested conclusion than a 10 days interval in the outside.

18. Line 361: How about the statistical indicators in per sub-figure? From Fig 7, the effect of piecewise fitting is significantly better than full fitting. The input of TGS2600 can be used to eliminate some predicted extreme outliers, but methane fitting results is not improved significant, especially at low concentrations.

    In Fig 4 data also show that a better resistance fit with the TGS2600 as baseline results with a resistance from 20 to 70. And it also just well confirms that the two TGS have similar changing characteristics potentially caused by temperature and $H_2O$. Based on the very weak methane fitting improvement performance after the input of TGS2600, the response of TGS2600 under low concentration conditions cannot be ruled out. Therefore, the input of TGS2600 is only more conducive to predicting the resistance of TGS2611, because their resistance changes in principle are relatively similar. Therefore, the experiment almost failed to achieve the purpose of improving methane retrieval by inputting TGS2600 to weaken the influence of other factors.

19. Line 371: Why not make the x and y axes change in the same range? This seems more intuitive.

20. Line 457, 'we did not find TGS2600 to respond to methane in the studied 2 to 10 ppm range.'. If this MS aimed to conclude this, and it requires more stronger evidence.

21. Line 458-459: 'We suggestion….'.  As comment 20 mentioned above, the response of TGS2600 to methane is not excluded in this MS and is therefore not recommend appear in here.

22. Line 461-463: The indoor and outdoor conclusions obtained by the author are not fair. The temperature and water changes are very small in the indoor experiment, which is a relatively ideal condition compared to the outdoor experiment. This ideal condition can significantly reduce the uncertainty caused by diurnal water and temperature in the baseline fitting (see comment 14). Moreover, correlation analysis also shows that resistance is related to temperature and water intensity. This implies that such networks have high challenging at outdoor application.

---

## Author Comment (AC1)

We appreciate the thorough, meticulous, and detailed review of our manuscript, and for the most part we agree with the reviewer's comments and have made revisions accordingly. In particular, the reviewer's focus on the manner in which we reported our regression results is thoughtful and helpful, and we are grateful to their contribution in improving our presentation of our work.

We respond to each comment below. For convenience, we reproduce the reviewer's comments in italics, and our responses are given in normal text.

*Dear Editor,*

*This manuscript tests a low-cost sensor node containing a Figaro TGS 2600 and a Figaro TGS 2611-E00 metal oxide sensor, alongside a high-precision reference instrument. The sensor was tested both outside and inside. The outside dataset could not be used due to a lack of methane enhancements, despite efforts to artificially enhance the background. The inside dataset could however be used. The correlation of both sensors to various environmental conditions and methane concentration were presented, using the TGS 2611-E00 as the main methane sensor. This work took the approach of deriving a baseline resistance corresponding to methane concentration below 2.3 ppm, to incorporate non-methane environmental effects, which included a time component. A second baseline was also devised which included TGS 2600 measurements, as this sensor was assumed not to be methane-sensitive. The ratio between measured resistance and baseline resistance compared to methane concentration was poor for both approaches. Instead, the baseline was split into 10 pieces for which individual baseline coefficients were derived. This approach was slightly better. A sensitivity analysis was conducted on piecewise baseline variability components over time. In another analysis, the TGS 2600 was also treated as the main methane sensor, but this did not yield results better than treating the TGS 2611-E00 as the main sensor. In any case, any analysis on methane concentration is slightly dubious as most methane concentration measurements were obtained in background conditions. The greater value of this work is in baseline resistance modelling. This paper is a welcome addition the growing body of literature on this subject. I was particularly impressed by the meticulous nature of this work. The standard of English and the presentation is very good. I suggest the following editions to the work, in addition to the two paragraphs below, where I address two key issues.*

*Using the TGS 2600 resistance measurements to derive a baseline fundamentally requires for TGS 2600 methane sensitivity to be negligible. The theory is that this (assumed) non-methane sensor can be used to characterise non-methane effects. Yet, if methane effects are not negligible, then methane sensitivity may be reduced in the resistance to baseline ratio, thus being counterproductive. Based on visual inspection of figure 7, there is not much improvement when comparing an eq 1 (no TGS 2600) baseline to an eq 2 (with TGS 2600) baseline versus methane concentration. The authors do not provide $R^2$ values for each fit and only provide values for their favoured fit (eq 2 piecewise). Unless the authors can prove that the eq 1 baseline approach is significantly better than the eq 2 baseline approach, I do not think using TGS 2600 measurements within a baseline is necessarily better. The authors' justification for using eq 2 instead of eq 1 is that it results in a better baseline fit (i.e. modelled baseline versus measured baseline). This is hardly surprising as this simply shows that two similar TGS units behave in a similar way. But the authors must also be sure that there is no TGS 2600 methane effect.*

*To test the TGS 2600 methane effect, the authors do apply a baseline correction to the TGS 2600 data (in the same way for the TGS 2611-E00 data), observing poor methane correlation. Yet, the authors fail to provide an $R^2$ value of the methane fit for comparison. Furthermore, most TGS 2611-E00 baseline attempts also failed, when compared to methane concentration. I think the 10 ppm limit and the number of data points may have been insufficient to reveal sufficient correlation for this sensor. I therefore question the conclusion from this manuscript to dismiss the utility of the TGS 2600 from further research. The authors must clearly state these caveats and limitations when discussing their views and outlook on the use of the TGS 2600. This work is not sufficient to conclude that the TGS 2600 is not sensitive to methane below 10 ppm.*

Thank you for your comments! We have added $R^2$ and RMSE values to the various fits to add context and support our claims. We believe that our claims about the unsuitability of TGS2600 for methane monitoring in this range are also supported by our previous laboratory work (Furuta et al., 2022), as we have stated in the manuscript. We have attempted to more clearly state the caveats and limitations in our revisions to your specific comments below.

**General comments**

*Abstract*
- *This is concise and easy to follow. However, this section is written slightly vaguely. The abstract should include more specific details on the work presented in the manuscript and not be afraid to use a bit of technical language. For example, a crucial advancement in this work is the development of a baseline resistance model. This should definitely be in the abstract. The authors could also add some more details on the enclosure (such as size), if appropriate.*
- *I also think that the duration of testing time should be included here. • I am interested by the decision not to specify that a Figaro TGS is being used either here or in the title. I do not insist that the authors do so. However, I would appreciate some rationale please, as I do not understand the value of this.*

We have added more details to the abstract as noted in our response to specific comments below, including specifying TGS models. We have attempted to balance detail and conciseness,

*Introduction*
- *This section provides a very good summary of recent research on the TGS in methane concentration measurements, with good referencing. However, this first paragraph could provide a better overall motivation for the necessity of this work in the context of increasing methane emissions, with some citations.*
- *The way in which the narrative of this section is conveyed is sometimes confusing as it is not clear whether it is written in the context of methane or more generally. Based on the first sentence, this is a paper on methane measurements. It should therefore be emphasised throughout this section that points are being made in the context of methane sensors and not more generally.*

Thank you for the comments. In the interest of not repeating the same content often found in

other papers, we have added a reference to a paper that effectively summarizes the motivations and applications for low-cost methane sensing. Please also see our responses to the specific comments.

*Methods*
- *Overall, this section is well-constructed and covers the methods very well. The calibration section was particularly well-written and easy to follow.*
- *Regarding the timestamps, was the lag time of the TGS enclosure ever tested? The 30 s lag time of the LI7810 was assumed to be negligible, which is fine. However, the lag time of the TGS logger is a bigger issue. If a spike of methane was emitted just beneath the TGS enclosure, I wonder how long it would take TGS resistance to peak in response. It would not be short (a few seconds), as air is carried to the sensors by diffusion. I do not think 10-minute averaging will overcome this problem, unless TGS peak time is less than 1 minute. The averaging is however still necessary to account for the nature of diffusion, which is not instantaneous like a LI7810. The overall solution is both a lag-time correction in combination with averaging to smooth any emission spikes.*

Thank you for your comments. We have added a figure to Appendix A illustrating the sensor lag with a series of large methane spikes - the sensor responds reasonably quickly to the spikes (within minutes), but takes considerably longer to decay back to baseline. It is not immediately clear to us that this can be accommodated through data processing, and we believe that the addition of a pump will likely be necessary to resolve this issue.

We have added a description of this lag to this section as well.

*Results*
- *The investigation of various causes of baseline disparity is a welcome inclusion in the research on this topic. The correlation plots and the baseline analysis are both good. However, the work on a methane response using different baseline approaches lacks detail.*
- *In addition, this section must include an analysis of methane correlation excluding periods used to produce a baseline fit. Otherwise, it is mostly an analysis of the ability of the baseline to predict the baseline as the vast majority of datapoints were below 2.3 ppm, i.e. the baseline threshold.*

Thank you for your comments. We believe we have addressed your points in our revisions to your specific comments below.

*Discussion*
- *The sensitivity analysis of the time component on the baseline is a very useful and robust way of identifying the nature of temporal variability.*
- *Testing the ability to derive a TGS 2600 baseline for methane correlation is a very good idea. However, the caveats must be discussed more clearly here. Furthermore, a plot of methane correlation would be welcome. It is not clear how much worse the TGS 2600 is than the TGS 2611-E00, when detecting methane.*

We believe we have addressed the caveats in our revisions to your specific comments below, and have added $R^2$ values to help give additional context to the relative performance of the sensors (we had initially omitted these as they are negative, which we thought might confuse

readers unfamiliar with the notation).

*Conclusion*
- *This is a nice overview of the work, but lacks a few key values and outcomes from the work, that could be included.*

Please see our revisions to the specific comments.

**Specific comments**

*Line 11: We deployed the prototype sensor alongside a reference methane analyzer in two sites - one outdoors, one indoors - for several months each of data collection across a range of environmental conditions and methane levels.*
- *This sentence is difficult to follow.*
- *Also, please specify which types of environmental conditions. Variations in temperature?*

Revised.

*Line 13: calibration models*
- *Please add a few very brief details on what sort of calibration models. What is the crucial basis of the calibration approach? Were linear models used? Was machine modelling used?*

Revised.

*Line 14: background monitoring and enhancement detection*
- *Monitoring and detection of what? Presumably this is methane concentration.*

Revised.

*Line 14: performance*
- *What does "performance" mean? Does this mean accuracy compared to the reference instrument? Please clarify.*

Revised.

*Line 14: 2 to 10 ppm range*
- *Please specify that this refers to methane concentration.*

Revised.

*Line 17: these and similar inexpensive MOx sensors*
- *Of which sensors? It is not specified which sensors have been used. The authors cannot refer to "these" sensors and compare them to similar sensors, if the sensor has not yet been introduced.*

Revised; we have added model numbers to the beginning of the abstract.

*Line 17: near-background methane monitoring*
- *What does this mean? Perhaps state that this refers to up to 1 ppm (or whatever) concentration enhancement.*

Reworded for clarity.

*Line 19: scarcity of high-resolution ground-level data*

*• Please provide some references for this statement.*

Although we believe this to be true, it is difficult to cite a lack, and an in-depth discussion would be tangential to our work; we have chosen to remove the statement. We agree with your comment.

*Line 21: variety of sensor mechanisms*
     *• Please specify that this refers to methane sensors. It is better to write this section in the context of methane detection, rather than general low-cost sensor use.*

Revised.

*Line 29: sensor array or "e-nose" configurations*
     *• Please briefly explain how using a sensor array can overcome issues of selectivity.*

Revised.

*Line 38: when calibrated in a laboratory setting*
     *• Please specific that it was both calibrated and tested in a laboratory setting. This 1.7 ppm value refers to an indoor laboratory test.*

Revised.

*Line 41: above 100 ppm concentrations*
     *• In the laboratory or in the field?*

Revised.

*Line 41: 2 to 100 ppm range*
     *• Again where? In the laboratory or at a glacier?*

Revised.

*Line 42: but found that quantitative emission estimates had poor accuracy • I don't understand the relevance of this. This is a manuscript on methane concentration measurements and not flux measurements.*

The cited study estimates fluxes from concentration estimates derived from TGS2611-E00 sensors, and so we believe there is some relevance; however, we agree with the reviewer's comments and have removed this clause to reduce confusion.

*Line 48: positive results*
     *• Please clarify, positive results in what? In methane concentration measurements? It is previously stated that this sensor can be used to measure many gases, so this must be clear.*

Revised.

*Line 40: Cho et al. (2022) find*
     *• Please write this in the past tense.*

*Line 48: Eugster and Kling (2012) report a*
     *• Write this in the past tense.*

*Line 49: Other papers find*

*• Write this in the past tense.*

Revised.

*Line 83: We burned in the sensors and regulator for a week prior to data collection. • Why*

Revised.

*Line 91: Our system performed well in both sections of the experiment, with 95% of all data showing a sensor supply voltage within ±0.25 mV of the mean and 99.99% of all data showing a supply voltage within ±0.80 mV of the mean across the full dataset.*
  *• This is a very good demonstration of good practice and an excellent result. It is impressive.*

Thank you for your comment! We hope that the system design given in the appendix will be helpful for future research.

*Line 119: Our first site was an urban yard in Minneapolis, USA*
  *• Perhaps provide the coordinates, if possible.*

Due to privacy concerns we are unable to provide the exact address; we have added the neighborhood, however, to give a better sense of the location.

*Line 125: The background methane concentration at our research site*
  *• How is this known? Is this based on the LI7810? Please specify.*

Revised.

*Line 127: 2.5% methane gas cylinder*
  *• What was the rest of the cylinder filled with Synthetic air? Natural air? Argon? Nitrogen?*

Revised; the cylinder was balanced with air, but we are unable to find further details on the composition (the manufacturer datasheet and the calibration certificate just say "Air" without specifying natural or synthetic).

*Line 129: These releases produced a maximum 10-minute averaged methane value of 5.8 ppm, with most of the releases producing methane concentrations between 3 and 4.5 ppm. • Again, please specify how this is measured. Is this based on the reference gas analyser?*

This is correct, revised.

*Line 133: Our second site was indoors in the Biosystems Engineering building at the University of Minnesota, Twin Cities campus.*
  *• This is a little confusing. Quite simply, one test was conducted was outside and one test was inside the building. However, it reads as if there were two totally different sites in different locations, which adds to the complexity. "Our second site" implies that there were two totally separate testing locations.*
  *• In reality, there was only one site, not a first and second site.*

Revised; we added context that the two sites are approximately 5 km apart.

*Line 136: We also expected possible emissions of methane and other gasses from surrounding labs*
 • *This could be a crucial point. The lab was filled with potential interfering gas sources, which could influence the TGS response. I think the authors may elaborate here on potential gases that could be present or, at least, provide some more specific details on the activities taking place in the vicinity.*

Revised to add some more context on possible things in the vicinity.

*Line 153: we removed two hours of data after each reboot*
 • *Can the authors confirm that 2 hours was sufficient for the sensor to stabilise after power loss? I suggest a plot showing this.*

We have added a figure to appendix A showing the startup behavior for each time the unit was powered up during the experiment. The curves are difficult to interpret, but it appears that the initial "warm-up" period for the sensors is rather short, possibly 10 minutes. Evaluating the full power-up behavior would be a useful contribution for future work.

*Line 156: dataset averaged to a 10-minute timescale*
 • *Was it a 10-minute running average or were 10-minute averages taken next to each other. Please clarify this both here and throughout the manuscript.*

Revised to clarify that we used consecutive 10-minute averages.

*Line 172: We chose to include sensor run time to incorporate any effects of sensor aging;* • *This is a very good idea. Is this the absolute time or the time of the sensor being switched on to account for data gaps?*

Revised to clarify that we used the cumulative time the sensor was turned on from the beginning of the experiment (the long data gap was due to an issue with the reference analyzer, and did not affect the sensor).

*Line 177: Relative humidity is dependent on water vapor concentrations and temperature* • *And also, pressure.*

Revised.

*Line 179: we decided a priori to include water vapor concentrations and not relative humidity as a possible term in our analysis.*
 • *I totally agree with this rationale. Perhaps cite some other research that preferred to use water concentration instead of relative humidity.*
 • *Another important detail is missing here. Is this water concentration from the LI7810 or is it derived from the SHT? If it is derived, the authors must state how it derived.*

Revised to clarify we used water vapor concentrations from the LI7810, and added a citation to Shah et al. (2023) which discusses this issue in greater depth.

*Line 193: The TGS2611-E00 sensor response is expected to deviate from the predicted baseline response as a result of methane levels as shown in Equation 3a.* • *Why was this equation chosen? It is based on any previous work? I have never seen a purely linear fit applied to a resistance ratio. This is no criticism, but I am interested to know where it comes*

*from.*

Previous work, such as Fig. 8 in Shah et al. (2023) or the curves in the sensor datasheets (e.g. Figaro USA, 2013), suggests a power function or logarithmic fit. However, this work also suggests that a fit in the limited methane range we exam may be approximately linear (similar to the small signal model in electronics work). We chose to use a linear equation for simplicity, and our results shown in Fig. 5 do not appear to suffer from bias as a result.

We have added a note that a power function will likely perform better over a wider range.

*Line 212: We observed a diurnal cycle in methane levels*
> *• How? Using the reference gas analyser?*

That's correct, revised.

*Line 213: soil processes*
> *• What does this mean?*

We have removed this as speculative.

*Line 240: humidity*
> *• What does this mean? Is this water concentration? The term "humidity" is used regularly throughout the manuscript. However, I do not think this is wise as it could mean either water concentration or relative humidity. I suggest that the authors replace all "humidity" terms in this manuscript with a more precise term (either water concentration, relative humidity or something else) to avoid ambiguity.*

This is an excellent suggestion. We have revised the paper throughout accordingly, replacing "humidity" with "water vapor concentration" in cases where it was ambiguous.

*Line 248: relative humidity*
> *• I'm not too sure in the value of including relative humidity here in this list and in the previous analysis. As the authors themselves state, relative humidity and water mole fraction are related. So, it is hardly surprising that both influence TGS resistance.*

This is correct. We chose to include relative humidity in the list and in the next sentence to remind the reader that we have chosen a priori to use water vapor concentration rather than relative humidity; we feel that restating this decision is helpful to the reader, but are open to revision if the reviewer feels strongly that it adds confusion.

*Line 252: by including TGS2600 in the baseline response, we can possibly remove influence from nontarget gasses and other unexpected factors.*
> *• The authors also remove the baseline from the TGS2600 dataset? This should be mentioned in this section somewhere.*

Our apologies, we're uncertain exactly what the reviewer is asking with this comment. We use the averaged TGS2600 in our baseline Equation 2 without removing data, if this is the question. We have added a citation to the previous sentence to support the claim that TGS2600 does not respond to methane in this concentration range.

*Line 260: 1.46, 1.56, and 2.81 kΩ*
> *• These values should include a plus-minus sign as the square root has a negative and positive solution. The same for RMSE below.*

We respectfully disagree with this suggestion. RMSE is equivalent to the standard deviation of the regression residuals, and accordingly should be represented as a positive magnitude (as the standard deviation conventionally is) rather than as a +/-. Furthermore, of the previous work in Atmospheric Measurement Techniques we cite, four papers (Eugster et al., 2020; Collier-Oxandale et al., 2018; Jørgensen et al., 2020; our previous work in Furuta et al., 2022) report RMSE as positive magnitudes, and one (Shah et al., 2023) reports +/- values. We accordingly prefer to keep RMSE values as positive numbers, both in keeping with the underlying statistical argument and in accordance with what appears to be typical convention.

*Line 293: The fit is the best quality*
   *• Which fit?*
Revised.

*Line 306: with R2=0.46 and RMSE=0.65 ppm*
   *• The authors should also provide RMSE and $R^2$ values for periods not used to derive a baseline (i.e. above 2.3 ppm). This would help to evaluate the capability of the approach when not applied to the same data used to derive a baseline fit.*
We agree with the reasoning of this suggestion. We have added $R^2$ values (0.39 with the outlier points, 0.58 excluding them) and RMSE values (0.73 and 0.59 ppm with and without outliers, respectively) above 2.3 ppm.

*Line 317: we calculated the absolute change in methane concentration with the data immediately before and after each point*
   *• What is the time distance between each point? Are they 10-minute running averages of averages next to each other?*
Revised.

*Line 338: occurrence after a data gap (D)*
   *• What are the red dots? I think this figure could be improved with the different colours given as a legend in each subplot. The figure caption is not clear as all of the points are plotted.*

*Line 371: Figure 7: Sensor calibration results using the different baseline regression approaches.*
   *• Please change the axis from "actual CH4" to "LI7810 CH4" or similar, as "actual" doesn't mean much.*
We appreciate your helpful comments on figure clarity. As reviewer 2 has also made some comments on the figures, we will work on improving the figures in our later response to their comments, with your suggestions in mind as well.

   *Line 439: However, neither possible fit (with the Equation 1 or Equation 2 baselines) for methane had R2 better than simply predicting the mean, nor RMSE better than 4.5 ppm. • Unless $R^2$ values are provided, corresponding to the plots in Figure 7, it is not clear whether the TGS 2600 is truly worse than the TGS 2611-E00. The non-piecewise baseline attempt for the TGS 2611-E00 also failed and only one model worked.*

We have added $R^2$ values for these regressions, as well as for the fits with different baselines in section 3.4. We had previously omitted these values out of concern that the reader would not know how to interpret negative $R^2$ values, but we agree with your point that specifying the values gives important context. We have attempted to explain that negative $R^2$ values indicate that predicting the data mean for all points is a better fit.

We believe that the drastically worse $R^2$ and RMSE values at this line as compared to those in section 3.4 helps support our argument that TGS2600 is a worse quality methane sensor at this low concentration range, an argument which is also supported by our previous laboratory work in Furuta et al. (2022).

*Line 452: parts cost of under $200*
> • *This is very impressive. It should be included in the methods section.* • *Also, please provide the cost of telemetry data transfer in the methods section. Presumably there is a regular subscription charge of some nature.*

Thank you! We have added the parts cost to the first paragraph of the methods section, and added a note that the Particle Boron module we used as a cellular modem came with an included data plan sufficient for our needs without extra or recurring cost.

*Line 457: We suggest that work finding TGS2600 to respond to methane in a low concentration range should consider possible co-emitted gasses, algorithmic overfitting, or other experimental factors.*
> • *I do not think that this sentence should be in the conclusion as they are not key findings from this work and are simply the opinion of the authors. We do not know for sure why the TGS2600 has worked in previous studies*

This is fair, we've removed this (and in the previous section as well).

*Line 462: above 10 ppm*
> • *Why above 10 ppm? This work was below 10 ppm. All available data above 10 ppm were excluded from the analysis so I do not understand why this point is being made.*

Reworded. We feel it is useful to give an opinion on the concentration ranges in which our sensor (or similar) may be relevant, and where it is likely not the best solution.

*Line 463: Our sensor response correlates with methane levels with moderate accuracy in the lower 2 to 10 ppm range*
> • *This is misleading for two reasons.*
> • *First, the authors must be precise what they mean by "moderate accuracy".* • *Second, most of the analysis was performed on data below 2.5 ppm. Only a small percentage of data was anywhere near 10 ppm. So, the sensor was mostly tested at background levels and not mostly over a range up to 10 ppm.*

As noted at the end of section 3.3, 45% of the inside data was 2.5 ppm or lower, while the majority was higher concentration - the data is certainly biased towards the low concentration range, but there was a range of concentrations represented. We also feel that this bias towards low concentrations most likely makes our results pessimistic - we would expect the sensor to perform better with a higher mean concentration.

We have revised the sentence to clarify "moderate" with RMSE.

*Line 474: controlling water vapor levels and temperature in the sensing chamber • Why would this help if the baseline model already accounts for these effects? Surely  the problem is unknown environmental factors. What is the use of controlling known variables?*

This is a good point; we have removed this clause.

*Line 476: methane concentrations above the near-background range*
    *• What does this mean?*
Revised to "above the 2 to 10 ppm range we examined."

We have also corrected a typo in the caption for Table B1 and have made some minor changes to wording throughout for clarity.

---

## Author Comment (AC2)

We appreciate the reviewer's thoughtful evaluation and critique of our paper. We generally agree with the reviewer's comments, and feel that the suggested revisions improve the clarity of the manuscript. We reply to the specific points below: the reviewer's comments are italicized and our responses are given in ordinary text.

1. *Line 14: 'moderate performance' could be expressed more quantitatively.*

   Revised to add an RMSE value of < 0.6 ppm.

2. *Line 19-20: It seems that the lack of high-resolution data and low-cost network have few causations. In addition, could you provide some stronger evidence to support the lack of high-resolution data?*

   Reviewer 1 commented on this as well; although we believe this to be true from observation, we agree that the assertion is not well supported and have decided to remove this statement.

3. *Line 75-76: Based on the previous experiments, it is a good idea to exclude the influence of environmental factors through the response difference between the two TGS sensors, thus enabling the monitoring of methane in low concentrations (1-10ppm). However, it is well known that TGS2600 is sensitive to hydrogen, ethanol, iso-butane, carbon monoxide (CO), methane, relative humidity (RH) and temperature, while TGS2611-E00 is only sensitive to methane, hydrogen, ethanol, RH and temperature owing to its additional filter covers. (see FIGARO TGS 2611-E00&2600 product information, the above factors are ranked in order of influence.).*
   *And therefore, does the effect of hydrogen, ethanol, isobutane, and CO be taken into account in evaluation of the application potential of the TGS2611-E00 in low concentration situation? Especially in the indoor food experiment room. The differences in principle caused by physical between the two sensors and the response characteristics of both two resistance should be presented here.*

This is an excellent point. We have added discussion of the specific gasses the two sensors are sensitive to, and explained how including both could help to account for some interfering gasses (hydrogen, ethanol), while other interfering gasses may cause a response in TGS2600 but not TGS2611-E00. It is also unclear to us which gasses were tested by the manufacturer, and accordingly if there are undescribed interfering gasses that might occur in our scenarios.

4. *Line 115: In Fig 4B, where is the indoor sample gas inlet of $CH_4$ reference instrument? It may affect time delay.*

   We added clarification in the main text that the reference analyzer was drawing air from directly next to the sensor node for the indoor site.

5. *Line 119: As mentioned above, TGS2600 is also responsible for hydrogen, ethanol, isobutane, and carbon monoxide, and therefore nearby sources of relevant gas emissions or its surrounding environment are needed to be additionally stated.*

   We agree; we have added that we are unaware of nearby sources of interfering gasses, in addition to methane.

6. *Line 122: Why these data are averaged at 10 mins scale rather other time scale? Is this a universal practice? or is it tested with some experiment then after 10 mins is selected? In addition, what do you means of 10 mins? Average all the data over 10 mins after then record a valid value?*

   We have expanded our mention of this towards the end of section 2.3 to explain that we averaged to 10 minutes to reduce the effect of lag for our passive sensor node, and to smooth out short spikes which might not diffuse quickly into our device. We choose the 10-minute scale at the beginning of our analysis as a

duration long enough to smooth out any very short transients, but short enough to have real-world utility. We have also added an explanation that the 10-minute average is consecutive averages of the collected data.

7. *Line 132: In the title of Sect 2.2.1, the author reports the situation of increasing background concentration, and how many ppm the concentration is it?*

We have added detail at the end of the section about the enhancements resulting from our controlled releases, specifically with concentrations measured by the reference analyzer. The highest 10-minute averaged value from a release was 5.8 ppm, with most releases falling between 3 and 4.5 ppm.

8. *Line 134-135: A brief description of laboratory-generated gases with potential effects on resistance is needed.*

We agree; we have added context that we expected VOCs, hydrogen sulfide, nitrous oxide, and other unknown gases. We have also added that the nearby lab conducts a range of bioprocessing projects, including manure processing and other fermentation studies.

9. *Line 173: 'time parameter'. Is the time parameter taken for the fit calculated from the start time of the per interval? In fact, within a 10-day window, environmental factors do not increase or decrease linearly but time does. Adding the time factor is a good attempt, but what is the contribution or significance of adding this factor?*

We agree this is ambiguous. We have added context that the time variable is the total time the sensor was powered on since the beginning of the experiment, which would account for sensor aging as well as environmental factors.

10. *Line 182: Is '2.3 ppm' the optimal selection after lots of tests? Or was it chosen randomly by the author? It would be an important impact factor on fitting result.*

We did not want to bias the selection by trying values and then choosing the best, and so we chose 2.3 ppm a priori based on our previous work as a threshold these sensors would likely not be able to differentiate from baseline. We have added text to this effect.

11. *Line 185: I have seen lots of fitting functions in Appendix B, but can you explain why log is used rather than others? Has any other research used a similar fitting formula before? Or is the log more appropriate after statistics? Some explanation is needed here.*

We have added brief additional discussion to Appendix B to note that we chose the log transform as a common statistical transformation. The regression residuals did not show clear bias, and so we did not investigate other transformations further.

12. *Line 212: Authors report that a diurnal cycle during rainy week is caused by soil process, please provide some solid evidence for this diurnal cycle driven by temperature.*

We have removed the mention of soil processes as speculative and have added an example of this diurnal pattern as an appendix.

13. *Line 277: Statistical indicators are recommended to be presented on the figure, and therefore looks more intuitive. In addition, the numbers of data on each sub-figure need to be represented. The first impression is that the numbers of data in ABCD is different from EFGH.*

We appreciate the suggestions on figure clarity. We agree with adding statistical results, and will label the panels more obviously.

*14. Line 281-283: 'Even with the additional sensor term, the accuracy of the regressions varies with time period, as can be seen in the coloring of Fig. 4. For example, the baseline at the beginning of the inside experiment in Fig. 4E has a worse fit than the baseline closer to the end of the experiment.'.*

*In Figure 2-A2, from early Apr to the end May, in this period temperature and $H_2O$ fluctuate over a wide range. While before 1 Apr, temperature and $H_2O$ changes obviously smaller, especially temperature. When the author only used only one piecewise to fit the whole study period, and the fitting result will obviously be affected by the number of samples. The former months are about 3 times than the latter months. Thus, it can be expected that the fit will be significantly worse in April and May, i.e. bring greater RMSE. And the data prefer to attribute this result to its representative rather than time.*

*In addition, in both Figures 4A and 4B, even if the author uses piecewise fitting in 4B, it can be clearly seen that the RMSE of the yellow points in both 4A and 4B is larger than the RMSE of the green points. Compared Fig-2A to Fig-2B, temperature and $H_2O$ in former months (Jan to Apr) also well show larger variation than latter months (after Apr) and therefore imply different possibility with author.*

This is an excellent point, and we agree that the increased fluctuation in environmental conditions is worth considering. However, the early period shows worse performance in Fig. 4E than the later months, which is the opposite of what we would expect if the increased environmental variability were solely responsible. As you say, in Fig. 4A the late period has a worse fit, as expected; so, it seems that some of the benefit of adding TGS2600 is improved tracking of environmental factors (as you have noted elsewhere). But, since the non-piecewise Equation 2 performs worse at the beginning of the experiment than at the end, despite the increased sample size of the relatively stable period of the experiment, we believe something else is going on.

15. *Line 293-295: I agree with baseline needs to be regression and is also a good experiment and perspectives. But the drive factor should be clarified.*

    To give a better sense of the importance of the different terms, we have added a sentence about statistical significance; the time term was non-significant in two of the ten piecewise subsets, and the intercept was non-significant in one, but all terms were otherwise significant in all subsets with p ≤ 0.001.

16. *Line 320: How about the averaged change speed of $CH_4$? Actually, I see lots of black point (high change speed points) on the red line.*

    We have added corresponding data on rates of change for the full dataset as follows: "...40% of the outliers show a rate of change greater than 1 ppm per 10 minutes; 31% exceed 2 ppm per 10 minutes; and 6% exceeded 5 ppm per 10 minutes. For the full dataset, 5.1%, 1.7%, and 0.16% show the same rates of change respectively, suggesting that the outliers are considerably more likely to occur during periods of rapidly changing concentrations than are the other data."

17. *Line 354-358: From Fig 2, the temperature and $H_2O$ in outside vary great in a day. And from Fig 3 data also Line 368 of MS show that resistance is sensitive to temperature and $H_2$ Why not have a try on constructing a fitting baseline by interval of temperature and $H_2O$? It might get more interested conclusion than a 10 days interval in the outside.*

    This is an interesting and reasonable suggestion; we have also found in our previous work that the sensor response varies by temperature and $H_2O$ bands (such as in Fig. 7 in https://amt.copernicus.org/articles/15/5117/2022/). However, the outside dataset in this manuscript has an extremely small methane range – almost all the data is 2 to 2.5 ppm – and we are unaware of plausible previous work suggesting the sensors will respond with sufficiently small error for this range. As our baseline fit for the outside data is already quite strong, we believe it is unlikely that an improved baseline fit will yield a better methane response; the fit for methane does not show any obvious trend, and so we conclude that the sensors

are simply unsuited for this low range without further breakthroughs, which we think are unlikely to be algorithmic.

18. *Line 361: How about the statistical indicators in per sub-figure? From Fig 7, the effect of piecewise fitting is significantly better than full fitting. The input of TGS2600 can be used to eliminate some predicted extreme outliers, but methane fitting results is not improved significant, especially at low concentrations.*
*In Fig 4 data also show that a better resistance fit with the TGS2600 as baseline results with a resistance from 20 to 70. And it also just well confirms that the two TGS have similar changing characteristics potentially caused by temperature and $H_2O$. Based on the very weak methane fitting improvement performance after the input of TGS2600, the response of TGS2600 under low concentration conditions cannot be ruled out. Therefore, the input of TGS2600 is only more conducive to predicting the resistance of TGS2611, because their resistance changes in principle are relatively similar. Therefore, the experiment almost failed to achieve the purpose of improving methane retrieval by inputting TGS2600 to weaken the influence of other factors.*

In consideration of this comment and comments from reviewer 1 we have added RMSE and $R^2$ values for the other three cases; we had previously omitted these as we were unsure if negative $R^2$ values (indicating performance worse than always predicting the data mean) would confuse the reader. All of the fits other than Fig. 7D produced negative $R^2$ values, although all clearly captured some methane trend. The fit using Equation 1, without TGS2600, appears to perform significantly worse than with TGS2600 included. It is unclear to us if this improvement is due to better tracking of T and $H_2O$, as you reasonably suggest, or if interfering gasses or some other environmental condition is responsible. Picking apart these different factors will require further research, and appears to us to be crucial for using these sensors in a low concentration range.

*19. Line 371: Why not make the x and y axes change in the same range? This seems more intuitive.*

We agree this will make the figure clearer, we will revise accordingly.

*20. Line 457, 'we did not find TGS2600 to respond to methane in the studied 2 to 10 ppm range.'. If this MS aimed to conclude this, and it requires more stronger evidence.*

*21. Line 458-459: 'We suggestion....'. As comment 20 mentioned above, the response of TGS2600 to methane is not excluded in this MS and is therefore not recommend appear in here.*

20 and 21: We agree with both you and reviewer 1 that our statement was stronger than the evidence supports. We have decided to remove the sentence beginning with "We suggest", and clarified that we did not find TGS2600 to respond in the range and with the algorithms we applied.

*22. Line 461-463: The indoor and outdoor conclusions obtained by the author are not fair. The temperature and water changes are very small in the indoor experiment, which is a relatively ideal condition compared to the outdoor experiment. This ideal condition can significantly reduce the uncertainty caused by diurnal water and temperature in the baseline fitting (see comment 14). Moreover, correlation analysis also shows that resistance is related to temperature and water intensity. This implies that such networks have high challenging at outdoor application.*

We agree; we have added a mention of the wider temperature and humidity range and noted that this is typical in outdoor settings in many climates.